# High Summertime Aerosol Organic Functional Group Concentrations from Marine and Seabird Sources at Ross Island, Antarctica, during AWARE

Jun Liu[1], Jeramy Dedrick[1,2], Lynn M. Russell[1], Gunnar I. Senum[3], Janek Uin[3], Chongai Kuang[3],
Stephen R. Springston[3], W. Richard Leaitch[4], Allison C. Aiken[5] and Dan Lubin[1]

[1]Scripps Institution of Oceanography, University of California, San Diego, 9500 Gilman Drive, La Jolla, CA 92093;
[2]Now at Texas A&M University, 400 Bizzell St, College Station, TX 77843
[3]Environmental & Climate Sciences Department, Brookhaven National Laboratory, Building 815-E, Upton, NY 11973-5000.
[4]Environment and Climate Change Canada (ECCC), Toronto, ON, Canada\
[5]Earth and Environmental Science, Earth Systems Observations, Los Alamos National Laboratory, Los Alamos, New Mexico, USA

*Correspondence to*: Lynn M. Russell (lmrussell@ucsd.edu)

**Abstract.**

Observations of the organic components of the natural aerosol are scarce in Antarctica, which limits our understanding of natural aerosols and their connection to seasonal and spatial patterns of cloud albedo in the region. From November 2015 to December 2016, the ARM West Antarctic Radiation Experiment (AWARE) measured submicron aerosol properties near McMurdo Station at the southern tip of Ross Island. Submicron organic mass (OM), particle number, and cloud condensation nuclei concentrations were higher in summer than other seasons. The measurements included a range of compositions and concentrations that likely reflected both local anthropogenic emissions and natural background sources. We isolated the natural organic components by separating a natural factor and a local combustion factor. The natural OM was 150 times higher in summer than in winter. The local anthropogenic emissions were not hygroscopic and had little contribution to the CCN concentrations. Natural sources that included marine sea spray and seabird emissions contributed 56% OM in summer but only 3% in winter. The natural OM had high hydroxyl group fraction (55%), 6% alkane, and 6% amine group mass, consistent with marine organic composition. In addition, the Fourier transform infrared (FTIR) spectra showed the natural sources of organic aerosol were characterized by amide group absorption, which may be from seabird populations. Carboxylic acid group contributions were high in summer and associated with natural sources, likely forming by secondary reactions.

## 1 Introduction

West Antarctica is one of the most rapidly warming regions on Earth (Bromwich et al., 2013), which has potential impacts for the melting of the Antarctic ice sheets and consequent sea level rise (Steig et al., 2009;Lambeck et al., 2002). In some regions, ambient aerosols contribute substantially to the radiation balance (Stocker et al., 2013), but little is known about the

sign and magnitude of their contribution in Antarctica because of the lack of measurements of their abundance, composition, and sources. In fact, there are few places on Earth where measurements of aerosols and their properties are needed to constrain modeled radiation as much as in Antarctica.

Since McMurdo Station is the only site with measurements of PM (Particulate Matter), Elemental Carbon, Organic Carbon,
and number concentrations that is within 300 km of the Ross Ice Shelf (which covers an area of more than 500,000 $km^2$). Furthermore, the station is unique in that McMurdo Station is one of the two sites that have published aerosol measurements starting in 1968, with the other one being the Amundsen Scott Station at the South Pole. The site has at least 10 publications describing aerosol measurements over the past 50 years, most of which were limited to summer (Cadle et al., 1968;Warburton, 1973;Ondov et al., 1973a;Hogan, 1975;Hofmann, 1988;Hansen et al., 2001;Mazzera et al., 2001a;Mazzera
et al., 2001b;Giordano et al., 2017;Kalnajs et al., 2013;Khan et al., 2018). No stations in Antarctica measured inorganic chemical composition year-round until 1978 (Parungo et al., 1981), and none have measured year-round organic components. In 1966, electron micrographs of particles collected on a four-stage impactor provided some of the first aerosol measurements carried out at McMurdo Station (Cadle et al., 1968). Filter samples were collected for elemental analysis in 1970-1971 (Ondov et al., 1973b). During the austral summers of 1969 and 1970, the Aitken nuclei concentration was
reported to be ~1000 $cm^{-3}$ (Warburton, 1973). In another study, the number concentration was 50 to 150 $cm^{-3}$ with continental winds and ~300 $cm^{-3}$ with maritime winds (Hogan, 1975). Balloon measurements were conducted later for stratospheric aerosols, and long distance signals from volcanic sources in tropical areas were found in the stratosphere (Hofmann et al., 1986;Solomon et al., 1994). Hansen et al. (2001) measured black carbon at McMurdo in austral summer in 1995-1996. Another study (Mazzera et al., 2001b) reported more detailed $PM_{10}$ elemental composition, elemental and
organic carbon, and nitrate concentrations for 1995–1996 and 1996–1997 at McMurdo. Chemical Mass Balance (CMB) receptor modeling estimated that soil dust, sea salt, combustion emissions, sulfates, methanesulfonate, and nitrates contributed 57%, 15%, 14%, 10%, 3%, and 1%, respectively, to the summertime $PM_{10}$ mass (Mazzera et al., 2001a). Kalnajs et al. (2013) showed that ozone depletion is correlated to aerosol concentrations because halogen-containing aerosol consumed ozone. An aerosol mass spectrometer (AMS) at a site 20 km northeast from McMurdo Station during October
2014 to December 2014 and August to October 2015 (Giordano et al., 2017) found sulfate accounted for more than 50% of non-refractory composition. Many measurement campaigns were limited to austral summer months because of restrictions on access (Cadle et al., 1968;Ondov et al., 1973a;Warburton, 1973) and so lack information on seasonal changes.

The few year-round aerosol concentration and composition measurements in Antarctica were collected at several sites in coastal Antarctica (all of which are more than 1500 km from McMurdo Station) (Hara et al., 2005;Wagenbach et al.,
1998;Jourdain and Legrand, 2002;Gras, 1993;Hara et al., 2004;Hara et al., 2010;Weller et al., 2013;Minikin et al., 1998;Read et al., 2008) and at several sites on the Antarctic Peninsula (more than 3000 km from McMurdo Station) (Asmi et al., 2018;Mishra et al., 2004;Kim et al., 2017;Saxena and Ruggiero, 1990;Savoie et al., 1993;Loureiro et al., 1992), as well as at the South Pole (more than 1000 km from McMurdo Station) (Hansen et al., 1988;Bodhaine et al., 1986;Harder et al.,

2000;Parungo et al., 1981;Bodhaine, 1983;Hogan and Barnard, 1978) and at Dome C (more than 1000 km from McMurdo Station) (Legrand et al., 2017b;Legrand et al., 2017a;Udisti et al., 2012). At the South Pole, aerosol particle number concentration ranged from 10 to 30 $cm^{-3}$ in winter and 100 to 300 $cm^{-3}$ in summer (Bodhaine, 1983;Parungo et al., 1981;Hogan and Barnard, 1978). This low winter and high summer seasonal difference has been observed also at coastal

Antarctic sites, but the average concentrations were typically higher with summertime concentrations ranging from 300 to 2000 $cm^{-3}$ and wintertime concentrations from 10 to 200 $cm^{-3}$ (Kim et al., 2017;Gras, 1993). Consistent with this seasonal difference in particle number concentrations, most summertime non-sea salt sulfate mass concentrations were at least 5 times higher than winter concentrations (Jourdain and Legrand, 2002;Weller and Wagenbach, 2007;Udisti et al., 2012;Legrand et al., 2017a;Asmi et al., 2018), likely because of the contributions from biogenic DMS emissions from the surrounding

Southern Ocean. However, most sea salt aerosols had wintertime maximum concentrations with more than two times more $Na^+$ mass concentrations in winter than summer (Parungo et al., 1981;Wagenbach et al., 1998;Jourdain and Legrand, 2002;Weller and Wagenbach, 2007;Jourdain et al., 2008;Udisti et al., 2012;Legrand et al., 2017b;Legrand et al., 2017a;Asmi et al., 2018).

The few hygroscopicity and CCN measurements reported near West Antarctica are also recent and sparse. DeFelice et al.

(1997) conducted CCN measurements at Palmer Station on the Antarctic Peninsula in January and February 1994. They collected CCN for 27 days at 0.3% and 1% SS and found CCN concentration to be between 79 and 158 $cm^{-3}$. Asmi et al. (2010) found that aerosol particles over the Southern Ocean are very hygroscopic with a growth factor of 1.75 at 90 nm. At King Sejong Station on King George Island, Kim et al. (2017) found that CCN concentrations are high in summer (~200 $cm^{-3}$) and low in winter (~50 $cm^{-3}$). Biological emissions from marine sulfate sources have been proposed to explain a large

fraction of CCN in the Southern Ocean region (McCoy et al., 2015). Biological sulfate aerosol accounts for 43–65% of the summer zonal mean CCN concentrations and 7–20% of the winter CCN over the oceans in the Southern Hemisphere, including the circumpolar Southern Ocean (Korhonen et al., 2008). This important role for biological sulfate in the Southern Ocean suggests that biogenic organic components may also contribute significantly to particle number and mass, but measurements of organic particles are too scarce to determine if this is the case (McCoy et al., 2015).

For comparison, in marine and Arctic regions, the organic composition of particles have shown a high fraction of hydroxyl group  (61% of OM for the North Atlantic and 47% of OM for the Arctic) as well as some alkane and amine groups, likely associated with sugars, carbohydrates, and amino sugars originated from biological materials in seawater (Hawkins and Russell, 2010;Modini et al., 2015;Russell et al., 2010;Frossard et al., 2013;Leaitch et al., 2017;Shaw et al., 2010).  Organic nitrogen has also been identified as a tracer component (0.02 to 10 ng $m^{-3}$) in aerosol particles in various studies in Antarctic

(Schmale et al., 2013;Barbaro et al., 2015b;Dall'Osto et al., 2017) and Arctic (Scalabrin et al., 2012;Dall'Osto et al., 2012) regions. Some of the few measurements of organic aerosol particle composition that have been made in marine and polar regions are those of amino acids, which are summarized in Table S1 (Mace et al., 2003a;Kuznetsova et al., 2005;Scalabrin et al., 2012;Barbaro et al., 2015b;Mace et al., 2003b;Wedyan and Preston, 2008;Shi et al., 2010;Matsumoto and Uematsu,

2005;Mandalakis et al., 2011;Violaki et al., 2010). Amino acids in remote marine and coastal regions have been used as markers for biological activities since they are natural chemical constituents of many marine and terrestrial organisms (Barbaro et al., 2015b;Scalabrin et al., 2012;Milne and Zika, 1993;Cowie and Hedges, 1992). In addition, amino acids contain organic nitrogen and specifically amine groups, which are also consistent with measurements in polar regions of

CHNO fragments (Schmale et al., 2013) and amine groups (Shaw et al., 2010;Frossard et al., 2011). Sugar, levoglucosan, phenols and anthropogenic persistent organic compounds were measured in ambient aerosols at Mario Zucchelli Station and Concordia Station (Zangrando et al., 2016;Barbaro et al., 2016;Barbaro et al., 2017;Barbaro et al., 2015a). Carboxylic acids with low molecular weights were also measured at Mario Zucchelli Station, Concordia Station, and Dumont d'Urville (Barbaro et al., 2017;Legrand et al., 2012).

The Ross Sea has a surprisingly high biological primary production rate in the summer, making it the most biologically active part of the southern polar region (Arrigo et al., 2008). Seabird emissions were linked to new particle formation (Weber et al., 1998) and to particles containing CHN and CHNO fragments (Schmale et al., 2013). The CHNO fragments identified by mass spectrometry have been associated with uric acid and other nitrogen containing components that are produced from penguin guano (Schmale et al., 2013). The ammonia emissions from seabird colonies have also been shown to contribute

substantially to atmospheric particle formation and cloud-albedo radiative effects in the Arctic (Croft et al., 2016b). Organic aerosol components were also associated with melt-water ponds in continental Antarctica (Kyro et al., 2013).

 AWARE (ARM West Antarctic Radiation Experiment) provides the most thorough yearlong aerosol and radiative property measurements yet obtained from Antarctica, and the only four-season time series of weekly FTIR measurements of organic functional groups in Antarctica. This manuscript characterizes the sources of organic aerosol across four seasons in

Antarctica. Dust, sea salt, and non-sea salt sulfate mass concentrations measured by XRF are used to separate the seasonal contributions to inorganic particle components.  Seasonal patterns of natural marine and coastal-sourced organic aerosol are identified from the functional groups after separation of local emissions.

**2 Methods**

The AWARE aerosol measurements were collected from 23 November 2015 to 29 December 2016 at the Cosray site on the

eastern edge of McMurdo Station (77.85°S, 166.66°E), which is located on the southern tip of Ross Island in Antarctica. To quantify seasonal differences, four seasons were defined as Summer (November through February), Fall (March through April), Winter (May through August) and Spring (September through October) (Figure 1). The four-month winter is characterized by irradiance of nearly zero and average temperature below -20 °C.  The four-month summer had irradiance above 250 W m$^{-2}$ and temperature higher than -10 °C. Spring and fall marked transitions between summer and winter. The

station hosts more than 1000 scientists and support personnel during austral summer and consumes more than 2 million gallons of AN-8 diesel fuel (with a 0.3% sulfur content by weight) for station operations (Mazzera et al., 2001a). The aerosol

inlet samples at ~10 m above ground level and has a rain guard and bug screen, 1000 L min$^{-1}$ turbulent flow through 4.6 m of large-diameter (20 cm ID), powder-coated aluminum tubing, a 2.1 m smaller-diameter tube (4.76 cm ID) that extracts 150 L min$^{-1}$ flow from the center of the larger-diameter tubing, and a flow distributor with five ports, each drawing 30 L min$^{-1}$ through 25 cm of 1.59 cm (5/8") inner diameter stainless-steel tubing. The size-dependent losses were measured below 10% for particles from 10 nm to 10 $\mu$m diameter (https://www.arm.gov/publications/tech_reports/doe-sc-arm-tr-191.pdf). Other details of the measurement system can be found online in the description of the second ARM Mobile Facility (AMF2, https://www.arm.gov/capabilities/observatories/amf) and Aerosol Observing System (AOS, https://www.arm.gov/capabilities/instruments/aos).

Ambient aerosol particles were measured by CPC (Condensation Particle Counter, TSI model 3772), HTDMA (Hygroscopic Tandem Differential Mobility Analyser, Brechtel model 3002), and CCN Counter (Cloud Condensation Nuclei, DMT model CCN100) and were collected on filters for off-line FTIR and X-ray fluorescence (XRF). CN (condensation nuclei from CPC) concentrations had frequent short-lived increases that typically had high concentrations (>1000 particles cm$^{-3}$ for 1 Hz CN), which we attributed to short-term local contamination events (SLCE) (Figure S1). High CN concentrations (>1000 cm$^{-3}$) occurred 48% of the time when the wind was from the west (Figure S2), which is the same direction as the McMurdo Station central facilities. However, westerly winds only occurred 3% of the time, so emissions at McMurdo Station were unlikely to account for most of the emissions. Spikes were separated using a "de-spike" algorithm based on running median filters (Beaton and Tukey, 1974;Tukey, 1977;Velleman, 1977;Goring and Nikora, 2002). We applied a running median length of 24 hr and weighted by cosine bell running mean of 24 hr to the 1 Hz CN concentration and assigned the CN concentration above the resulting filter as SLCE. The SLCE were characterized by an average duration of less than 1 hr (0.5 min$\pm$6 min), rapid rate of concentration change (8520$\pm$36780 cm$^{-3}$ min$^{-1}$), and concentrations exceeding 1000 cm$^{-3}$. After SLCE (spikes) were removed, the 24-hr running median concentration was interpreted to be the natural background CN, for reasons discussed in Section 3.

Submicron aerosol particle samples were collected on pre-scanned Teflon filters (Teflon, Pall Life Science Inc., 37 mm diameter, 1.0 $\mu$m pore size) behind a PM$_1$ sharp-cut cyclone (SCC2.229 PM$_1$, BGI Inc.). One sample filter and one background filter were collected each week. Samples were frozen and transported to the UCSD laboratory for FTIR spectroscopy. A Bruker Tensor 27 FTIR spectrometer with a deuterated triglycine sulfate (DTGS) detector (Bruker, Waltham, MA) was used to scan the filters both before and after sampling. An automated algorithm was applied to quantify the mass of the organic functional groups (Takahama et al., 2013;Russell et al., 2009). Four groups (alkane, amine, hydroxyl and carboxylic acid) were quantified by the area of absorption peaks and the sum of the mass of the five functional groups. Other groups (organonitrate, organosulfate and non-acid carbonyl) were fit but all samples were below detection limit. The detection limit and error for each functional group is the larger of twice the standard deviation of the absorption values associated with blank filters and the visual determination of the minimum peak size that could be distinguished from spectral noise (Maria et al., 2002). The detection limit of OM was 0.09 $\mu$g based on the sum of the detection limits of the three largest

functional groups during the project (alkane, hydroxyl and amine). For the weekly air sampling volume of 80 m$^3$ used in this study, this loading corresponds to a concentration of 0.001 μg m$^{-3}$. OM is calculated as the sum of all functional groups measured above detection, based on the assumptions of Russell (2003). Subsequent evaluations and intercomparisons (Takahama et al., 2013;Russell et al., 2009;Maria et al., 2002) have shown that errors associated with functional groups that

are not quantified because of Teflon interference and semivolatile properties are accounted for within the stated ±20% uncertainty for ambient particle compositions. The ammonium mass is not quantified by FTIR of Teflon filter samples because ammonium nitrate is semi-volatile. The location of absorption by sulfate in FTIR coincides with the location of Teflon absorption. Since the absorption by the Teflon filter far exceeds that of the sulfate particles, sulfate cannot be measured on this substrate. Sulfur was measured by XRF and is expected to be largely ammonium sulfate, since

organosulfate and bisulfate were below the limit of quantification. Pure (>99%) uric acid (Sigma-Aldrich) and urea (Fisher Scientific) were dissolved in water, atomized and collected on triplicate Teflon filters to provide FTIR reference spectra for comparison of the amide group region. FTIR spectra were baselined by subtracting a combination of piecewise linear and polynomial regressions from the spectrum using an automated algorithm (Takahama et al., 2013).

Positive Matrix Factorization (PMF) was applied to the baselined FTIR spectra for the PM$_1$ samples collected in 2016 at

McMurdo Station with PMF2 V4.2 (Paatero and Tapper, 1994;Paatero, 1997). Six-factor solution spaces (1~6) were considered.  Fpeak values from -2 to 2 at 0.5 increments were considered. Seeds of 1, 10 and 100 were used at each Fpeak and factor number to examine the robustness of each solution. There was little change in solutions with rotations for all solutions. Q/Qexpected decreases as factor number increases for all solutions (Table S2). The two-factor solution is considered robust because the spectra are almost identical for all rotations and seeding conditions (Figure S3). The solution

leaves an average of 23% of the OM as residual. The two factors are not correlated in time and do not have similar spectra (Table S2). The new factor identified from the 3-factor solutions is either degenerate or very similar (cosine similarity =0.99) to one of the first two factors. Similarly for 4 or more factor solutions two or more degenerate or duplicate factors are found. This makes the two-factor solution with Fpeak of 0 optimal for the AWARE data set. The small number of factors identified compared to other regions (Russell et al., 2011) is the result of both the low aerosol concentrations and limited personnel

access at AWARE, which reduced the time resolution of FTIR samples to one week each and yielded only 54 samples in one year. The low variability during the study also meant that PMF was unable to separate more than two factors.

In addition, K-means clustering (Hartigan and Wong, 1979) was applied to the baselined FTIR spectra (Takahama et al., 2013). Solutions with 1 to 10 clusters were evaluated. The 2-cluster solution was chosen because solutions with 3 or more clusters included at least one pair of clusters with centroids with cosine similarity higher than 0.95 (Table S2), making those

clusters effectively overlapping. The two clusters and two PMF factors were identified as associated with Fossil Fuel Combustion (FFC) and Marine and Seabird (M&S) sources, as described below. Factorization techniques like PMF are applied to separate each individual composition measurement into the independent factors that contribute to its composition, where these factors may represent different sources as well as different formation processes. On the other hand, clustering

algorithms are used to sort similar measurements into categories, each of which may contain a mixture of different sources and formation processes and is characterized by the centroid to which all measurements in that category are most similar. The similarity of the k-means centroids and PMF factors (cosine similarity > 0.97) indicates that both separations are robust. Since the PMF residual is the fraction of OM that could not be assigned to either factor, the ratio of the residual to the factor

OM provides a measure of the uncertainty of the PMF separation – namely the fraction of OM that could be missing from the factor. The ratio of the PMF residual to the FFC OM varies from 29% in winter to 63% in summer, making this result more likely to represent all of the FFC OM in winter when FFC OM is a larger relative fraction of OM. Similarly, the PMF residual is 33% of M&S OM in summer, indicating the source separation could be missing a third of M&S OM.  In contrast, the PMF residual is 9 times larger than the M&S OM in winter (Table 1), making the quantification of M&S OM in winter

very uncertain.

Half of the filters (25) were selected for X-ray fluorescence (XRF) (Chester Labnet, OR) quantification of major elements above 23 amu. The elements Na, Mg, Al, Si, P, S, Cl, K, Ca, Ti, V, Cr, Mn, Fe, Co, Ni, Cu, Zn, Br, Rb, Sr, Zr, Ag, Pb and Ba had mass above detection limit (3 times the uncertainty) for 95% of the samples and are used here. The mass of dust was calculated from XRF metal concentrations, assuming dust consists of $MgCO_3$, $Al_2O_3$, $SiO_2$, $K_2O$, $CaCO_3$, $TiO_2$, $Fe_2O_3$, MnO

and BaO (Usher et al., 2003) after excluding mass associated with sea salt. Sea salt particle mass components were calculated from XRF-measured Na and Cl concentration (Frossard et al., 2014b;Modini et al., 2015).

The CPC measured particles with diameters larger than 10 nm and operated continuously, except from 29 March to 7 April 2016 when a malfunction occurred (Figure S1). The CCN Counter measured the particle concentration activated at supersaturations of 0.1%, 0.2%, 0.5%, 0.8%, and 1.0% during AWARE, with only short time periods of missing data (Figure

S1). HTDMA provided humidified aerosol size distributions for five dry particle sizes at specified relative humidity (RH = 90%) for two periods during the campaign: 23 November to 20 December 2015 and 16 to 31 January 2016. Aerosol particle growth factors ($GF_i$) from the HTDMA measurements were calculated as the ratio of humidified particle diameter of size i to the selected dry diameter. Mean growth factors (GF) and hygroscopicity parameters ($\kappa$) (Petters and Kreidenweis, 2007;Su et al., 2010) were calculated from Eq. (1) and Eq. (2):

$$\overline{GF} = \frac{\sum_i GF_i (\frac{dN}{dlogD_p})_i}{\sum_i (\frac{dN}{dlogD_p})_i} \tag{1}$$

$$\kappa = \frac{(\overline{GF}^3 - 1)(1 - a_w)}{a_w}, \tag{2}$$

where N is the measured number concentration and $a_w$ is water activity (Rickards et al., 2013).

Meteorological variables (temperature, humidity, wind speed and wind direction) were measured with a Vaisala model WXT-520 (Helsinki, Finland). The Surface Energy Balance System (SEBS) included upwelling and downwelling solar and

infrared radiometers at the measurement site at McMurdo Station from 4 February to 29 December 2016. Aerosol absorption

was measured at three wavelengths (470, 522 and 660 nm) by a Particle Soot Absorption Photometer (PSAP; Radiance Research, Seattle, WA). The PSAP absorption at 660 nm was used as a proxy for black carbon (BC) because it is expected to have the least interference from brown carbon (Olson et al., 2015).

### 3 CN, CCN, Hygroscopicity, and Inorganic Particle Measurements

19% of the 1-Hz CN measurements recorded during the project were identified as SLCE, and the average of the concentrations for those times contributed 55% of the project-average CN concentrations. The distribution of SLCE duration and timing (Figure S4) shows that SLCE events were approximately two times more frequent during local daytime than nighttime. This short duration and largely daytime timing of SLCE suggests that site maintenance and nearby road traffic are likely responsible for many of the high CN events.

There are two reasons why the CN concentrations that remain after SLCE (spikes) are removed are considered representative of the natural background rather than local pollution from McMurdo Station activities: First, the SLCE CN concentration is correlated weakly to BC (r=0.48), but the background CN is correlated negatively to BC absorption (r=-0.4). Second, the two indicators of combustion-related pollution (BC absorption and the FFC factor) were approximately two times higher in summer than winter (Table 1), which is similar to the two-fold increase in SLCE CN in summer compared to winter but not

enough to account for the seven-fold increase in the background (SLCE-removed) CN in summer compared to winter. Consequently, this larger summertime difference in background CN is likely associated with the higher productivity of natural sources in summer. More specifically, the CN concentration associated with natural sources was very low ($\sim$60 cm$^{-3}$) in winter during low phytoplankton activity but as high as 2000 cm$^{-3}$ in summer (Figure S1), indicating a significant increase in biogenic (sulfate or organic) CN.

SLCE had nearly no contribution to CCN, which is consistent with SLCE particles being extremely low hygroscopicity and freshly emitted from fuel combustion (Wex et al., 2010) (Figure S1).  The CCN measurements did not have short-term spikes even at the highest supersaturation level (1%), at which only 0.1% of the measurements were 5% higher than the background CN. The absence of the SLCE in the CCN measurements is likely the result of the local pollution being both too small and too low hygroscopicity to serve as CCN at 1% or below. The CCN concentration correlated moderately or strongly

to background CN (r=0.80, 0.83, 0.87 and 0.88 for 0.2%, 0.5%, 0.8% and 1% SS, respectively). CCN and CN were 5 to 7 times higher during summer, but the ratio of CCN/CN changed less than 30% throughout the year (Table 1). CCN/CN was largely constant at all five supersaturations during most of 2016, but from late September to early October the ratio of CCN/CN decreased to 0.5 at 1% supersaturation (Figure S1). This decrease of the ratio of CCN to background (spike-removed) CN during the winter-spring transition could be caused by changes in particle size and composition.  One such

cause would be additional CN that are too small to contribute to CCN. Previous observations at a site 10 km from McMurdo Station showed an increase in the fraction of CN smaller than 250 nm at polar sunrise (September-October), although a

specific cause was not clear (Giordano et al., 2017). The higher CCN/CN ratio in the summer (Table 1) is consistent with both the higher biogenic sulfate contributions during the highest productivity season (summer) and the slightly larger diameter of the accumulation mode particles observed in previous summers (Kim et al., 2017).

The growth factors and hygroscopicity parameters were both nearly constant during the two measurement periods (Figure S5), with values of 1.5±0.3 for growth factors and 0.4±0.1 for hygroscopicity parameters. These numbers were constant across the measured size range of 50 nm to 250 nm diameter and are comparable to other observations in the Antarctic region (Wex et al., 2010;Asmi et al., 2010;Kim et al., 2017). The particles that had too low hygroscopicity to grow measurably may be those that were emitted by local anthropogenic emissions. The moderate correlation of BC absorption to the fraction of particles that did not grow at increased relative humidity in the HTDMA (R=0.52, Figure 2 (a)) indicates that the BC-containing particles could be the particles that have low hygroscopicity. In addition, BC absorption correlated moderately to the non-activated CN particles (1-CCN/CN) (R=0.34 for 1% supersaturation, Figure 2 (b)). Since BC-containing particles, such as those freshly emitted from combustion sources, have been shown to have low hygroscopicity (Peng et al., 2017; Vu et al., 2017), these correlations are consistent with the particles that did not take up water being those that were emitted by local combustion activities.

XRF measurements of elemental concentrations of S, P, K, Ca, Si, Mn, Al, Ag, Fe, and V were 2 to 15 times higher in summer than in winter (Figure S6). Submicron dust mass concentration was 7 times higher in summer, consistent with the lack of exposed soil in winter (Figure 1). Sea salt particle mass concentration (Figure 1) was 3 times higher in winter than in summer, consistent with the higher circumpolar wind speed providing more sea spray in winter than summer (Bintanja et al., 2014). The measured $Cl^-/Na^+$ of 2 represents a large sodium deficiency in wintertime submicron particles (Figure 1). The depletion of $Na^+$ relative to $Cl^-$ in winter indicates a likely contribution to the aerosol submicron mass from wind-blown frost flowers (Alvarez-Aviles et al., 2008; Thomas and Dieckmann, 2003; Stein and MacDonald, 2004; Papadimitriou et al., 2007; Giannelli et al., 2001; Belzile et al., 2002; Shaw et al., 2010). This sodium depletion is the result of $Na_2SO_4$ precipitating out from sea ice brine before frost flowers wick up the remaining salt solution. Blowing snow could also contribute to submicron particles (Domine et al., 2004), but this source has not been associated with a substantial sodium deficiency in submicron particle composition (Gordon and Taylor, 2009).

If either frost flowers or blowing snow were generated near the site, we would expect a correlation of concentrations to wind speed at higher wind speeds, since both sources have been characterized as requiring wind speed thresholds of approximately 7 m s$^{-1}$ for lofting of particles (Schmidt, 1981;Shaw et al., 2010). During AWARE, 1-min wind speed only exceeded this threshold by 1 m s$^{-1}$ for 24% of the time, and the weekly average wind speed was never higher than 7 m s$^{-1}$. Wind speed had no correlation to CN concentration for the campaign (r=-0.32) or for winter (r=-0.31). In addition, there was no correlation (R=-0.15) of submicron CN number with wind speed (>8m s$^{-1}$), as would be expected for blowing snow generated locally (Yang et al., 2008). The M&S factor concentration also showed no correlation (r=0.1) to the fraction of time with high wind speed (>8 m s$^{-1}$). While these relationships do not support the attribution of the wintertime salt mass to either frost flowers or

blowing snow, they do not rule it out since the particles may have been lofted upwind and transported to McMurdo Station. A recent model simulation (Huang and Jaegle, 2017) predicted that blowing snow has significantly higher contributions to submicron particle mass than frost flowers in Antarctica and the Arctic, but also showed that the region at the north edge of the Ross Ice Shelf (including Ross Island) had both higher emissions ($>0.6 \ 10^{-6}$ kg m$^{-2}$ d$^{-1}$) and concentration ($>1.5$ µg m$^{-3}$)

from frost flowers than the emissions ($<0.4 \ 10^{-6}$ kg m$^{-2}$ d$^{-1}$) and concentration ($<1.0$ µg m$^{-3}$) from blowing snow, consistent with the finding that wintertime OM at McMurdo Station were more likely from frost flowers than blowing snow.

## 4 Organic Mass and Composition

The measured organic functional group mass concentrations are shown in Figure 3(c). The average OM is 0.13 $\mu$g m$^{-3}$ for AWARE, with hydroxyl groups having the highest mass fraction (41%), followed by alkane (39%), amine (13%) and

carboxylic acid (7%) groups. Similar to CN concentrations, OM was highest in summer (0.27 µg m$^{-3}$) and lowest in winter (0.04 µg m$^{-3}$). Arctic OM at Barrow and Alert showed a very different seasonal pattern with low concentrations in Arctic summer (0.03 µg m$^{-3}$ and $<0.5$ µg m$^{-3}$ in Alert and Barrow, respectively) and high concentrations in winter and spring (0.3 µg m$^{-3}$ and 1 µg m$^{-3}$ in Alert and Barrow, respectively) (Frossard et al., 2011;Leaitch et al., 2017). Consistent with OM, CN concentrations at these two Arctic sites, with particle size range of 80-500 nm at Alert and $>100$ nm at Barrow, were also

low in Arctic summer ($<50$ cm$^{-3}$ and 100-300 cm$^{-3}$ at Alert and Barrow, respectively) and high in winter and spring ($>100$ cm$^{-3}$ and 400-1000 cm$^{-3}$ at Alert and Barrow, respectively) (Croft et al., 2016a;Polissar et al., 1999). The springtime high concentrations in the Arctic result from long-range transport from mid latitudes after the breakup of the vortex. The lack of substantial pollution sources at southern mid-latitudes (compared to those at northern mid-latitudes) means the Antarctic does not have an equivalent haze in spring (Stohl, 2006;Stohl and Sodemann, 2010;Russell and Shaw, 2015). The higher

summer OM in Antarctica is likely produced by the specific local conditions of the three polar sites, namely Ross Island has higher marine and seabird activity compared to Barrow and Alert.

The FFC cluster and factor are similar to each other (cosine similarity=0.97) and are both named because of the similarity of the spectra to factors identified as FFC previously (Price et al., 2017;Guzman-Morales et al., 2014;Saliba et al., 2017). The FFC Factor has two narrow peaks at 2865 and 2934 cm$^{-1}$ that are characteristic of long-chain hydrocarbons and a cosine

similiarity greater than 0.8 with factor spectra identified previously as urban combustion emissions (Guzman-Morales et al., 2014) and fresh ship engine emissions (Price et al., 2017). The FFC factor has alkane and amine groups that account for 80% OM (Figure 4), which is consistent with urban combustion emissions and vehicle engine tests (Guzman-Morales et al., 2014;Saliba et al., 2017). The FFC factor was 73% OM in winter but only 23% in summer (Figure 3 (a) and (b)). The FFC factor concentration is weakly or moderately correlated to Ca, P, Fe, Cu, Cr, Mn and Zn (r=0.3~0.5), which have been

identified as tracers of vehicle emissions (Lin et al., 2015;Cheung et al., 2010).

The primary amine peak (1620 cm$^{-1}$) is present in both FFC and M&S factors at McMurdo Station (Figure 5), consistent with previous studies (Shaw et al., 2010;Guzman-Morales et al., 2014;Price et al., 2017;Leaitch et al., 2017). The difference between the FFC and M&S spectra is that FFC has double sharp alkane group peaks at 3000 cm$^{-1}$ but M&S has a broad hydroxyl group absorption at 3400 cm$^{-1}$ (Figure 4). Ammonium has peaks at 3050 and 3200 cm$^{-1}$ and contributes to both

FFC and M&S spectra (Figure 4).

The M&S Factor is identified as "marine" because of its high hydroxyl group fraction, which is similar to past marine sea spray factors (Russell et al., 2010), and as "seabird" because of absorption from ammonium and an organic nitrogen peak that is likely associated with coastal penguin emissions. The high hydroxyl group that accounted for 55% OM in the M&S factor makes this factor overall similar to the marine factors identified in measurements at Barrow and Alert (cosine

similarity=0.53-0.57) (Shaw et al., 2010;Leaitch et al., 2017) (Figure 3 and Figure 4). The M&S hydroxyl group fraction is lower than the Arctic marine factors that have 80% hydroxyl group (Shaw et al., 2010;Leaitch et al., 2017).

Barrow and Alert had higher marine OM concentrations in winter than in summer. Likely this is because these two Arctic sites did not have the large seabird contributions that contributed to the M&S factor on Ross Island during summer (Lyver et al., 2014). The smaller seabird populations near the Arctic sites also meant that Barrow and Alert OM had only very small

amide contributions (Figure 5). The M&S factor has higher alkane (38%) and amine (8%) group mass compared to two marine factors in Arctic regions that had only 6% alkane and 6% amine group mass (Shaw et al., 2010;Leaitch et al., 2017). This factor contributed a substantial fraction of organic mass in summer (58%) but very little in winter (5%) (Figure 3(b)). The M&S organic mass concentration was only 0.001 $\mu$g m$^{-3}$ during winter and was 0.15 $\mu$g m$^{-3}$ during summer (Figure 3(d)). The low winter and high summer M&S OM means that salt was not correlated to the M&S Factor organic mass,

indicating the high summertime concentrations of natural OM could not be explained by primary marine aerosol contributions alone. Marine OM contributions could be high in winter relative to summer because of the higher regional wind speeds, but their absolute concentration was too low to separate and identify in this set of 54 one-week samples. Specifically, the small number of long-duration samples resulted in PMF residuals that were more than 9 times higher than the M&S factor in winter, so that the marine fraction in winter is very uncertain.

The FTIR spectra for summer samples show an absorption peak at 1680 cm$^{-1}$ that is not present in winter (Figure 1). The M&S factor FTIR absorption peak (Figure 5) was located at a wavenumber that was both too high (>1630 cm$^{-1}$) to be primary amine bending and too low (<1714 cm$^{-1}$) to be carbonyl bending (Figure 5) (Takahama et al., 2013). Seabirds excrete urea that degrades to uric acid, and the amide groups found in both urea and uric acid could explain the 1680 cm$^{-1}$ peak in the summer FTIR spectra (Figure 5). The ammonium peaks (Figure 4) associated with the M&S factor are also

consistent with ammonia emissions from guano (Legrand et al., 1998), which is taken up on particles as ammonium.

More than 155,000 breeding pairs reside in the ice-free areas on Ross Island (Attwood et al., 2014) from October to March (Davis et al., 2001). The three penguin habitats on Ross Island are all less than 100 km from McMurdo Station (Figure S2)

(Lyver et al., 2014). Previous studies have also attributed aerosol emissions and properties to penguin activities, including ammonia-enhanced new particle formation (Weber et al., 1998) and oxalate-enriched particles and organonitrogen-containing fragments from urea breakdown products (Legrand et al., 2012;Schmale et al., 2013). The finding here of amide groups would be consistent both with particle formation and with substantial organonitrogen components. Since McMurdo

Station is most frequently downwind from Cape Crozier (which is located to the northeast of the sampling site), its estimated ~300,000 penguins are a likely source of this organic and ammonium contribution to particles (Lyver et al., 2014).

This 1680 cm$^{-1}$ amide peak was present in very small amounts in multi-year Arctic FTIR measurements (Shaw et al., 2010;Leaitch et al., 2017) (Figure 5), but their low concentrations did not support further investigation. The 1680 cm$^{-1}$ peak has not been observed in open ocean marine factors (Russell et al., 2010;Frossard et al., 2014a), suggesting that an open

ocean marine source is not likely. An alternative explanation of the amide group is emissions from seasonal ice microbiota (Dall'Osto et al., 2017). Given the proximity and abundance of seabirds at McMurdo Station, seabirds are the more likely source than are sea ice algae or other phytoplankton during AWARE. There are four reasons that the M&S factor are likely associated with marine and seabird emissions: The 1680 cm$^{-1}$ signal has been found at two coastal Arctic sites (in small amounts) but not on open ocean marine studies (Hawkins and Russell, 2010;Leaitch et al., 2017;Shaw et al., 2010;Frossard

et al., 2011). This difference suggests that the amide group is likely associated with seabirds, since they are found in coastal marine areas but generally not in open ocean marine areas. The higher concentrations of the M&S OM factor coincided with the summer breeding period of a large penguin colony at Cape Crozier, which was upwind during most of the summer. Other possible contributions, such as from algal blooms during ice melting in spring, are not consistent with the northeasterly winds, the amide group, or the seasonality of the M&S OM. HYSPLIT back trajectories (Draxier and Hess, 1998) did not

add useful information because the day-to-day variability exceeded the differences among weekly averages. Weekly-average wind direction was always northeasterly ($\pm$45 degrees), so there was insufficient variation to identify sources in different directions. The emissions from seabirds have significant regional implications in polar areas because of their large population and wide distribution (Croft et al., 2016b;Riddick et al., 2012). Chemical transport model simulations suggest that emissions of reduced nitrogen from seabirds in the Arctic could significantly increase aerosol particle formation, and in turn

cloud droplet number concentration and cloud albedo, yielding as much as -0.5 W m$^{-2}$ radiative forcing averaged over the 14,000,000 km$^2$ of the Arctic Ocean (Croft et al., 2016b).

The measured acid group concentration is likely to be a secondary aerosol contribution since photochemical oxidation has been shown to form highly oxidized molecules including carboxylic acids by photochemical reactions (Xu et al., 2013;Barbaro et al., 2017;Kawamura and Gagosian, 1987;Sax et al., 2005;Charbouillot et al., 2012;Alves and Pio,

2005;Claeys et al., 2007;Alfarra et al., 2006;Stephanou and Stratigakis, 1993). Acids are also present in trace amounts in seawater (Gagosian and Stuermer, 1977;Kawamura and Gagosian, 1987), but the higher concentrations measured here are likely to only be explained by secondary processes. The carboxylic acid group mass concentration that was associated with the M&S factor was correlated moderately to downwelling shortwave irradiance (r=0.75, Figure 6).

Carboxylic acid group mass fractions have also been identified as secondary photochemical products based on their correlation to solar radiation in clean, open-ocean conditions (Frossard et al., 2014a). However, since the seabird emissions were only high in summer when radiation was also generally high, the correlation to radiation does not provide evidence of photochemical contributions in this case. Interestingly, the carboxylic acid group associated with the FFC factor had no correlation (r= 0.09) to downwelling shortwave irradiance. This difference may be because the local emissions from McMurdo Station facilities reached the Cosray site in less than 5 min (since McMurdo Station was 2 km away and wind speeds were 6 m s$^{-1}$ on average) making them essentially "fresh" primary particles, whereas those from the large upwind penguin colony took 6 hr (since Cape Crozier was 100 km away and wind speeds were 6 m s$^{-1}$ on average) to reach the site giving them approximately 50 times more time for photochemical reactions leading to SOA production. It is also possible that the anthropogenic gas-phase precursor emissions had lower SOA acid yields but there is little evidence to support this (Rickard et al., 2010;Wyche et al., 2009;McNeill, 2015). The source of the vapor-phase organic precursors of the summer seabird acid groups is not known, but given their substantial contribution to mass is worthy of further investigation.

## 5 Conclusions

The first year-long organic functional group measurements in Antarctica show the seasonal trend of higher summer concentrations in most of the aerosol measurements. Short-lived contamination events (SLCE) of typically less than 1 hr (Figure S1) from local sources were separated from the CN time series to investigate the more regionally-representative or "background" concentrations. With SLCE removed, average CN concentrations were 65 cm$^{-3}$ in winter but 400 cm$^{-3}$ in summer.

The ratio of CCN to background (spike-removed) CN was largely constant for most of the measured seasons. Growth factors (1.5±0.3) and hygroscopicity parameters $\kappa$ (0.4±0.1) were measured in two one-month periods during the 2015-2016 summer and are comparable to marine aerosols reported near Antarctica (Wex et al., 2010; Asmi et al., 2010; Kim et al., 2017).

Both natural dust and biogenic as well as anthropogenic concentrations were more abundant in the summer months due to both the higher sunlight for productivity and the higher site accessibility. The mean summer OM concentration was 0.27 $\mu$g m$^{-3}$, which was 7 times higher than winter OM. Hydroxyl and alkane groups were found to be the most abundant and accounted for 80% of OM. Two factors were identified by PMF with an average residual of 23%: the M&S factor was associated with natural marine sea spray and coastal seabird sources, and the FFC was associated with local combustion emissions. The M&S factor mass concentration was 150 times higher in summer than winter; the FFC factor had a higher concentration than M&S in winter but the concentrations were so low that the quantification of the M&S factor in winter is very uncertain.

In addition to the primary amine peak present in past marine sea spray measurements, an FTIR absorption peak at 1680 cm[-1] was associated with the M&S factor in summer. The likely source of this peak as well as the coincident ammonium concentrations was seabird-related emissions from penguin colonies at Cape Crozier. The carboxylic acid group mass in the M&S factor was high in summer and was likely from secondary products of photochemical reactions.

5   **Acknowledgement**

We would like to thank the ARM (Atmospheric Radiation Measurement) program for the AMF-2 AWARE campaign, which was jointly supported by National Science Foundation (NSF AWARE grant: DPP-1443549) and Department of Energy (DOE Award number: DE-SC0017981.)  We also thank the AWARE personnel, Ryan C. Scott, Colin Jenkinson, Heath H. Powers, Maciej Ryczek, and Gregory Stone, for help collecting samples on site and Savannah Lewis and Gary Cheng for

10   assistance with sample preparation and analysis at Scripps. FTIR and XRF measurements are available at UCSD digital archives: https://doi.org/10.6075/J0WM1BKV. Other data are available on the ARM Data Discovery: https://www.arm.gov/research/campaigns/amf2015aware

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

**Table 1. Mean concentrations and ratios with standard deviations during 2016 at McMurdo.**

| Season | | Spring | Summer | Fall | Winter |
|---|---|---|---|---|---|
| CCN Number Concentration cm$^{-3}$ (CCN/SLCE-Removed CN) | 0.1% SS | 11.2±13.3 | 40.1±34.2 | 9.7±6.6 | 7.1±8.5 |
| | | (0.07±0.06) | (0.08±0.06) | (0.06±0.05) | (0.1±0.09) |
| | 0.2% SS | 37.9±36.4 | 131±80.2 | 48.2±29.3 | 18.6±20.5 |
| | | (0.19±0.11) | (0.26±0.12) | (0.29±0.11) | (0.26±0.14) |
| | 0.5% SS | 72.1±48.5 | 276.4±147.9 | 104±60.7 | 33.3±25.3 |
| | | (0.37±0.20) | (0.56±0.24) | (0.63±0.20) | (0.49±0.26) |
| | 0.8% SS | 99.7±73.9 | 348±203 | 124±72.3 | 42.9±39.6 |
| | | (0.5±0.23) | (0.68±0.25) | (0.75±0.23) | (0.57±0.29) |
| | 1% SS | 117±110 | 371±234 | 132±77.5 | 48.5±50.2 |
| | | (0.55±0.24) | (0.73±0.26) | (0.8±0.23) | (0.6±0.30) |
| CN cm$^{-3}$ | CN SLCE-Removed | 161±94 | 400±228 | 141±88 | 65±77 |
| | CN | 376±571 | 740±693 | 241±187 | 237±502 |
| Absorption mM$^{-1}$ | | 0.2±0.47 | 0.34±0.66 | 0.16±0.66 | 0.2±0.50 |
| Measured FTIR OM μg m$^{-3}$ | | 0.06±0.04 | 0.27±0.16 | 0.07±0.06 | 0.04±0.02 |
| PMF of FTIR OM | FFC OM μg m$^{-3}$ | 0.03±0.01 | 0.06±0.05 | 0.03±0.02 | 0.03±0.02 |
| | M&S OM μg m$^{-3}$ | 0.018±0.028 | 0.155±0.121 | 0.026±0.046 | 0.001±0.001 |
| | Residual/FFC | 0.40±0.72 | 0.63±0.84 | 0.36±0.49 | 0.28±0.52 |
| | Residual/M&S | 1.12±097 | 0.33±0.46 | 1.03±0.63 | 9.22±7.74 |

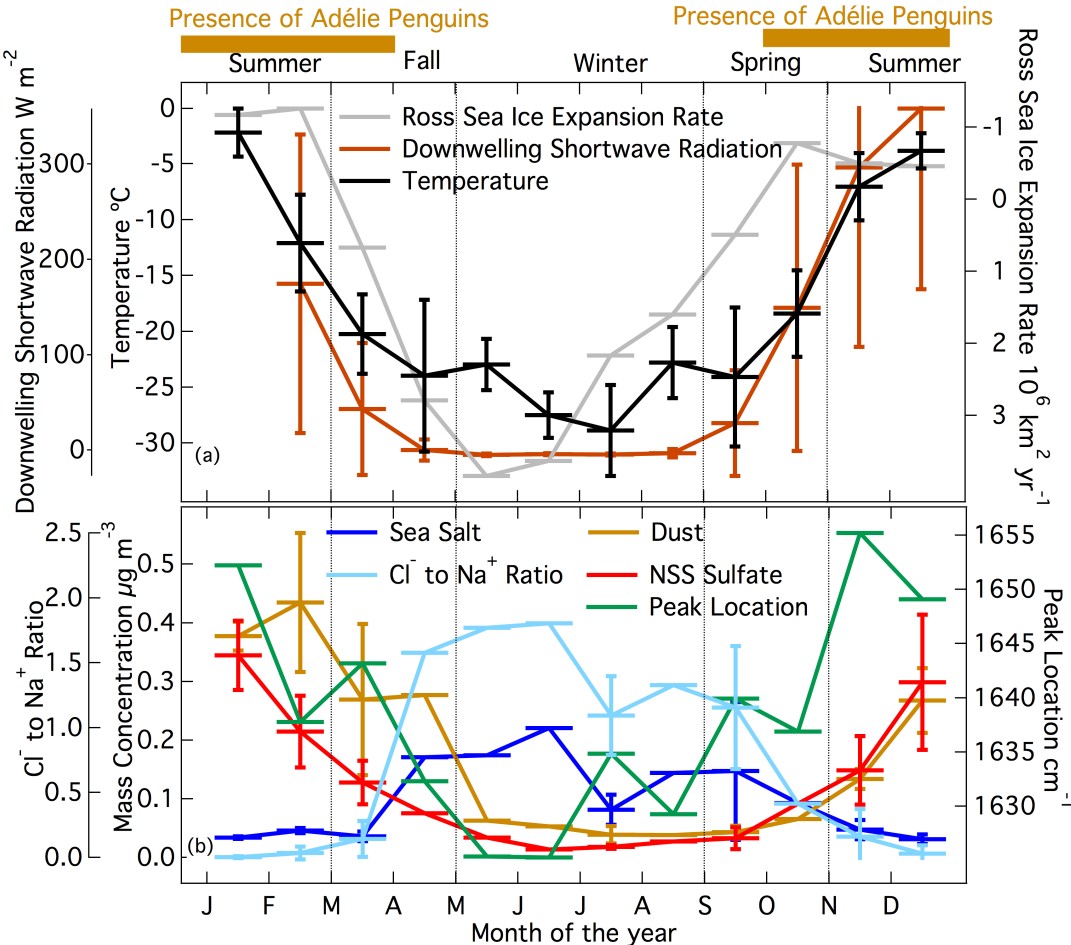

**Figure 1. Monthly average of (a) Temperature, shortwave downwelling irradiance measured in this study and sea ice expansion rate of the Ross Sea (Holland, 2014); (b) Sea salt, dust and non-sea salt sulfate concentration from XRF and FTIR peak location at 1500~1800 cm⁻¹ wavenumber region. Standard deviations are shown on the plot as error bars.**

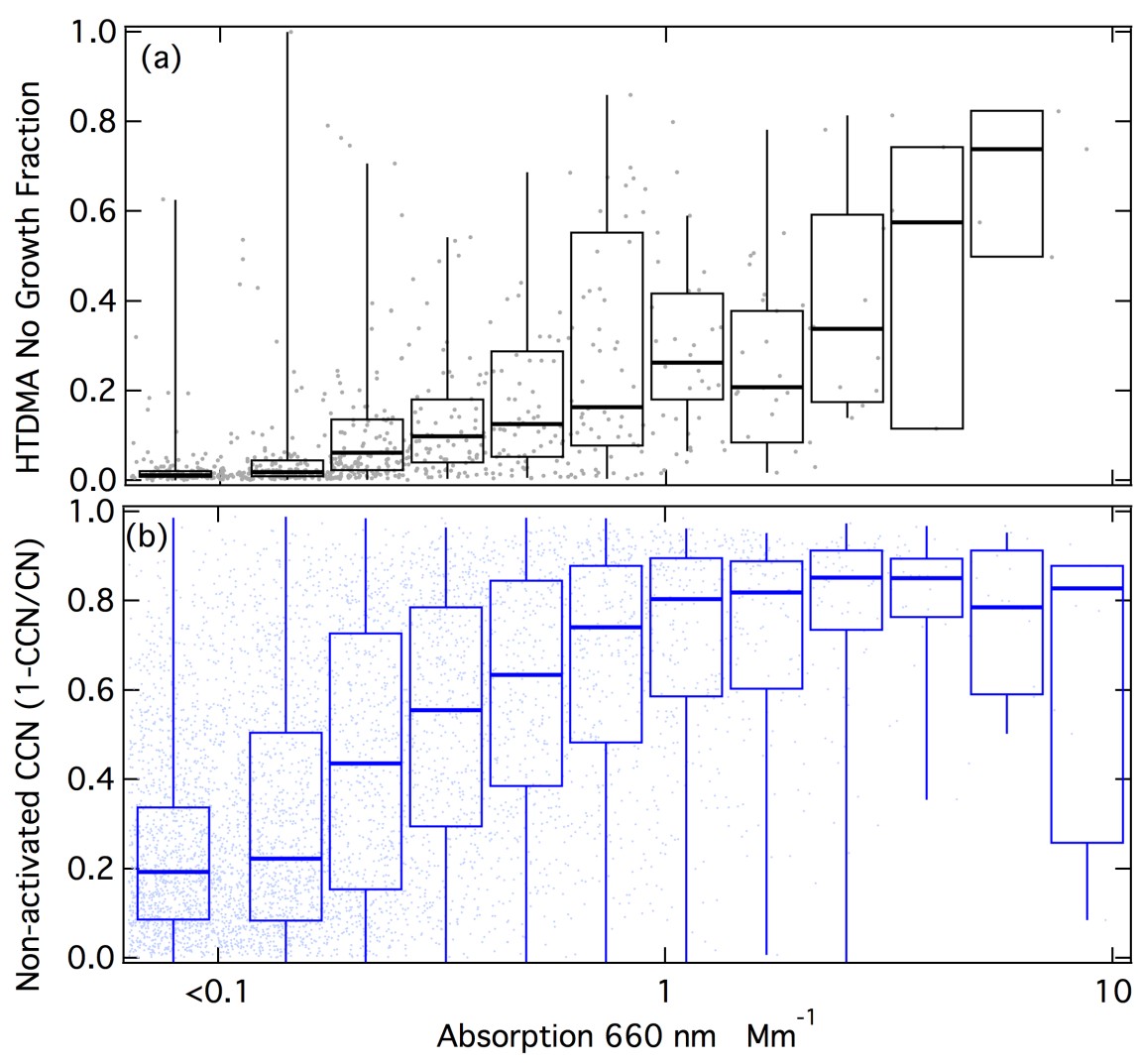

**Figure 2.** Scatter and box-whisker plot of PSAP 660 nm absorption and: (a) HTDMA no growth fration (r=0.52); (b) Non-activated CCN fraction (1-CCN/SLCE-removed CN) for 1% supersatuartion (r=0.34). The boxes show the 25th, 50th and 75th percentile values; the Whiskers show the minimum and maximum values.

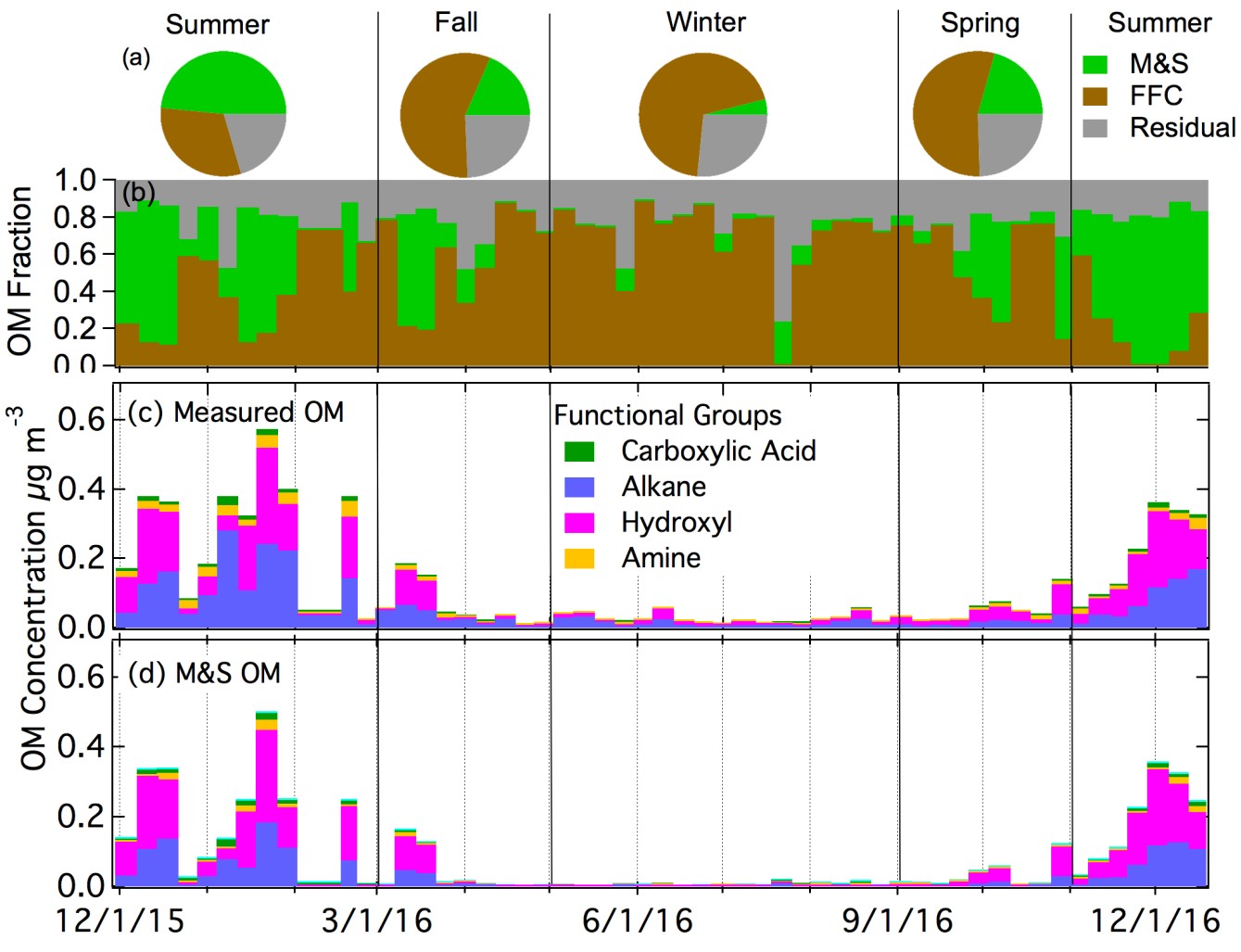

**Figure 3. (a)** Mass fraction of PMF factors in four seasons. Time series of **(b)** PMF factor OM fractions, **(c)** OM concentration with functional groups and **(d)** M&S OM concentration with functional groups.

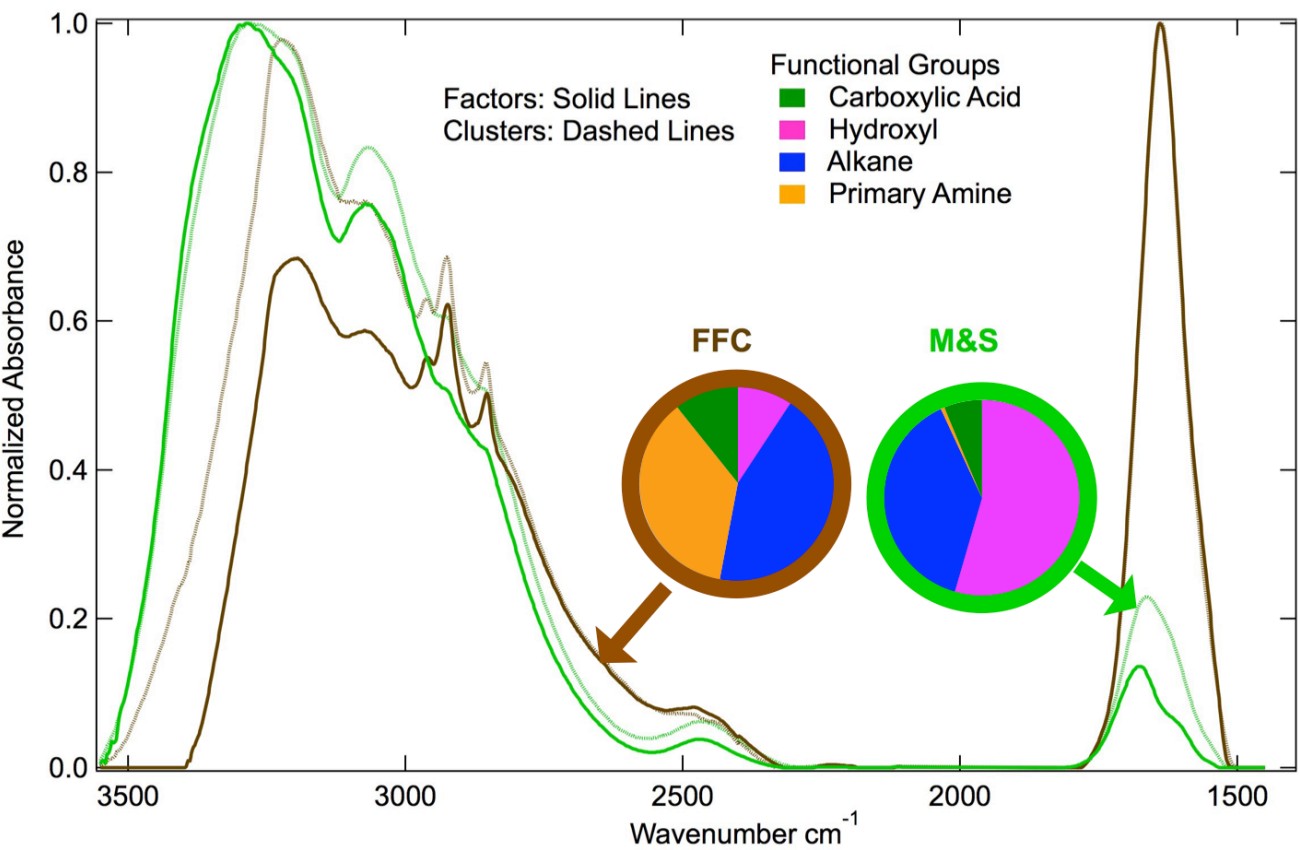

**Figure 4. Normalized spectra from k-means clustering centroids and PMF factors. Functional group fractions of PMF factors are shown in the pie charts.**

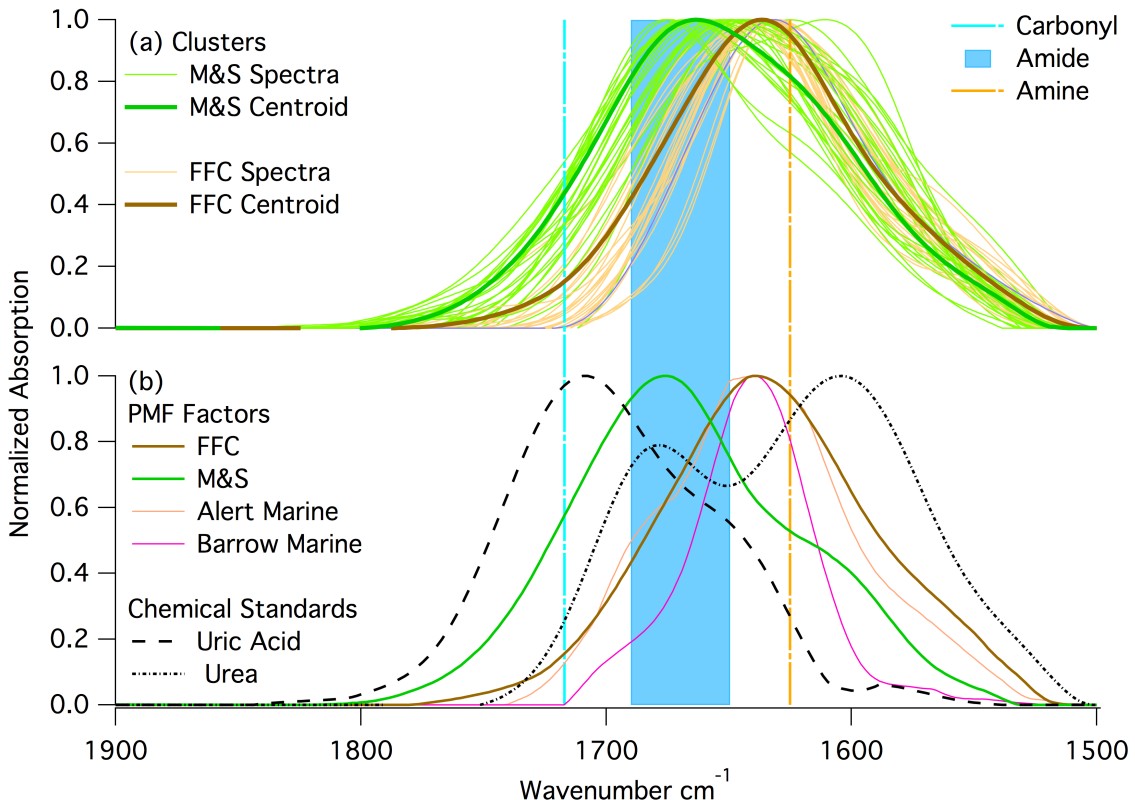

**Figure 5. Normalized spectra at 1500 to 1800 cm$^{-1}$ wavenumber region from (a) K-means clustering centroid and spectra in the clusters; (b) PMF factors from this study and two previous arctic studies, and chemical standards: urea and uric acid. Locations of primary amine and carbonyl group are marked on the figure.**

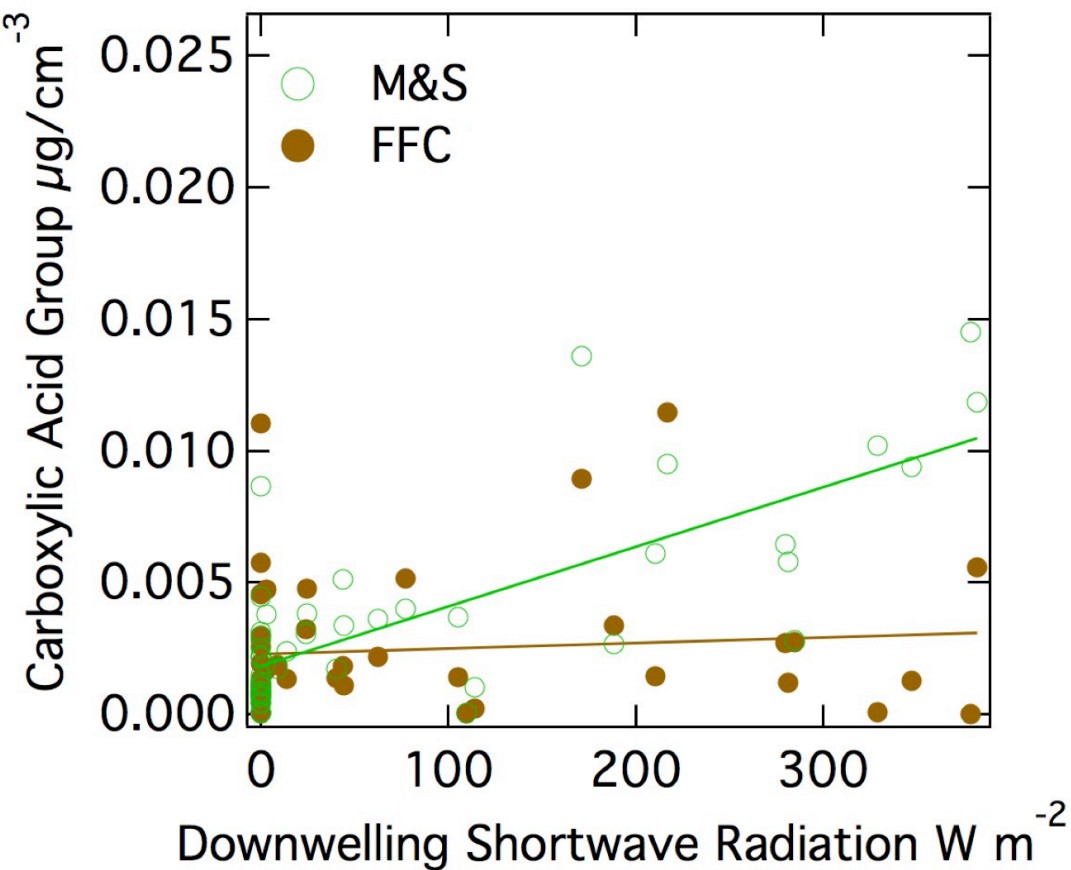

**Figure 6. Scatter plot of (a) M&S carboxylic acid group and shortwave downwelling radiation (r=0.75) and (b) carboxylic acid group in FFC and shortwave downwelling radiation (r=0.09)**

