# Peer review of "High Summertime Aerosol Organic Functional Group Concentrations from Marine and Seabird Sources at Ross Island, Antarctica, during AWARE"

_Atmospheric Chemistry and Physics, 2017_

## Referee Comment (RC1) · Anonymous Referee #1 · 24 Jan 2018

**General Comments:**

This manuscript provides an overview of one year of measurements of aerosol number concentrations, cloud condensation nuclei, and organic mass and functional groups (by FT-IR) made near McMurdo Station at the southern end of Ross Island in Antarctica from November 2015 to December 2016. The authors characterize the sources of organic aerosol, and are able to establish some evidence for the contribution of both primary marine organic aerosol and the formation of secondary organic aerosol. Both the seasonal coverage of these measurements and the general lack of measurements in this region makes these measurements very valuable and deserving of publication

in ACP if the following issues can be addressed.

**Major Comments:**

1) This manuscript generally lacks clarity in the text and is difficult to follow. Many paragraphs contain a number of seemingly disparate thoughts or concepts. I suggest that the authors give this manuscript a very thorough revision with particular attention to the structure of the text.

2) The scientific questions being addressed in this work need some clarification. I agree completely with the author's statement on P3L2-4 that this is most thorough yearlong aerosol data set from Antarctica. However, the questions that are addressed with this unique dataset are not clear, especially at the end of the introduction. After reaching the end of the manuscript it was more clear that the authors aimed to (1) characterize the sources of aerosol across seasons in Antarctica - especially of organic aerosol - and (2) establish evidence for the contribution of primary versus secondary formation of this organic aerosol. This could be stated more clearly at the end of the introduction.

3) Related to point (2), above, the authors spend a considerable amount of the text describing how the local pollution aerosol was removed from the data set. Once this aerosol signal has been removed (only from the total CN measurements, as far as I can tell) the authors then state (P5) that the remaining aerosol is from natural sources. It is very difficult to make this statement without first walking the reader through all the evidence you have for this point (e.g., how can you be sure that the "background" concentrations you measure are not impacted by local pollution). Further, the authors use a "despike" algorithm to remove high values of CN from the total particle concentrations, but do not filter any of the other data (i.e., CCN concentrations). Do you obtain similar results if you use a wind-sector and wind-speed based filter to remove contamination from local pollution sources? How does a wind-sector filter impact CCN concentrations?

4) The Results and Discussion sections (Section 3 and 4) seem to provide only very cursory discussions of the results. These sections also contain some topics (e.g., PMF details including seeds and Fpeaks) that are better placed in the methods section. Further, the authors seem to assume a significant amount of prior knowledge about the methods and measurement location, all information needed to understand and interpret these measurements should appear in the paper or the supplement.

5) The attribution of the "Marine and Seabird" PMF factor is not very well discussed. I do not dispute the claim, as it does appear likely given the proximity of large penguin colonies. Something as simple as a wind direction analysis relating high concentrations of the M&S factor with winds from the direction of penguin colonies would help your argument. Providing an analysis of back trajectories is also advisable.

**Minor Comments:**

P1 L30-31: This line reads as though the authors are implicating aerosol to some extent in the melting of the Antarctic ice sheet. I do not dispute that there can be a connection between atmospheric composition and ice dynamics (e.g., through cloud processes), but these processes are not nearly as well understood as the authors seem to suggest here.

P2 L1-21: This paragraph reads as a list of what has been done by others in this region. I agree this such a summary is an important part of this manuscript, but the introduction generally lacks a take-away message. For example, do the authors believe that we can say from past measurements that regional marine biological emissions are an important contributor to Antarctic aerosol? Do we know more about the biological sources of sulfate than organic aerosol? How does this help to motivate your study?

P2 L27 (and Table S1): What do we learn from this collection of amino acid measurements?

P3 L 11-20: Rather than referring to a web link for further information on the sampling

site, and pertinent information should be summarized here. Information about inlets? Length, flow rates, transmission efficiency as a function of size?

P4 L3: The detection limit needs to be defined in order for the statement "above the detection limit" to convey any information.

P4 L29: CN has not been defined at this point - I assume this is the CPC measurement?

P5 L1-22: The topic of this paragraph is not clear. There is a mid-paragraph shift from discussing pollution aerosol and its removal from the data set to discussing "background" aerosol.

P5 L3: This "despike" algorithm need to be described in more detail in the methods section. Since the goal of this paper is likely not to demonstrate the effectiveness of this "despike" algorithm, this isn't really a result but rather a part of data quality control that is applied before any interpretation can be made.

P5 L5: SLCE needs to be defined.

P5 L5-6: What does "accounted for" mean in this context? You need to provide a more quantitative assessment to make this statement. Further it is unclear that is meant by "occurred 19% of the time."

P5 L7: A mean $\pm$ standard deviation doesn't really tell you that the change was rapid in time, a rate of change might be a better metric.

P5 L11: Is this the author's hypothesis? If so, it should be stated as such. At this point in the analysis, it seems difficult to make this statement. For example, the remaining CN could be from background pollution not immediately associated with plumes.

P5 L13: "...correlated with..."

P5 L19 (Figure S1): Actually plotting CCN/CN would be informative, rather than asking the reader to estimate this ratio by looking at separate plots of CCN and CN

P5 L19-22: The meaning in these two sentences is difficult to discern. My sense is that the authors are trying to make a statement about changing CCN activation diameters, but it is unclear (i.e., what "decrease in particle diameter"? Associated with what phenomena?). Could the change in CCN/CN ratio result from either a change in particle diameter in summer or a change in composition that results in a change in the activation diameter?

P5 L23-33: This paragraph begins by discussing aerosol growth factors and ends by discussing aerosol absorption. The connection the authors are trying to make, if any, in unclear.

P5 27-29: The meaning of this sentence is unclear (i.e., a much smaller change in what?).

P6 L2: Figure 2 is referred to in the text before Figure 1.

P6 L3-6: Is the sea salt measured at this cite during winter transported form open water areas? What evidence do the authors have for this? The authors go on to suggest that frost flowers are the major source of sea salt at this site, but more discussion is warranted here. While, sodium to chloride or sodium to sulfate ratios provide some evidence, I do not see that this rules out other sources of aerosol, such as blowing snow, which has been observed in Antartica (e.g., Jones et al., 2009 www.atmos-chem-phys.net/9/4639/2009/). Field observations and laboratory experiments seem to suggest that frost flowers are rigid and difficult to break, even at wind speeds up to 12 m/s (e.g., Yang et al., 2017 https://doi.org/10.5194/acp-17-6291-2017; Roscoe et al., 2011 doi:10.1029/2010JD015144). A recent model study suggests that the frost flower source of sea salt aerosol cannot explain the seasonality of sea salt aerosol across several Arctic stations (Huang et al., 2017 doi:10.5194/acp-17-3699-2017). A more thorough discussion of possible sources supported by meteorological variables and back trajectories is needed to draw a conclusion here, especially since upward migration of brine and incorporation of frost flowers can lead to depletion of the sulfate-

to-sodium ratio relative to bulk sea water such that this chemical signature may not be unique to frost flowers.

P6 L11: Define what "baselined" means in this particular case. This should probably be discussed in the methods section.

P6 L13-15: These specific belong in the methods section. In the Results and Discussion section I suggest you discuss why your choice of two factors is the physically most meaningful choice (some of this discussion exists, but could be expanded).

P6 L 24-25: The motivation for carrying out a PMF analysis and a k-means cluster analysis of the FTIR spectra is not adequately explained here. Presumably if these two approached yield similar results, then this lends confidence to assignment of source types. This information needs to be stated clearly.

P6 L25-26: This needs to be actually shown in some way, even in the supplement.

P6 L32 (Table 1): In addition to stating the observation that the residual is much larger than the MS factor in winter, the authors should discuss what is means for interpretation of their results. This comment applies to all "ratios" provided in Table 1, the authors state that this provides a measure of uncertainty, how do the authors want the reader to interpret this? For example, does this result suggest that the PMF factors are robust across seasons? Only in specific seasons? This information should be clear.

P7 L11: Is the FTIR method sensitive to inorganic ammonium? If so, can a spectrum of e.g., ammonium sulfate be included to support this hypothesis.

P7 L15: "80 % hydroxyl group" - Is this a fraction of the total organic mass? How do you know that all organic mass is accounted for? Do you have to assume an average molecular weight for aerosol components? More description of how you arrive at these values is warranted (in the methods section).

P7 L15-16: Are these differences between Arctic and Antarctic summer aerosol significant? Do you expect a difference? More discussion is needed here, rather than just

stating that there is a difference between these two, arguably rather different, regions.

P7 L19: What is you detection limit for total organic mass?

P7 L19: Is it not the other way around - high summer OM and low winter OM attributed to marine and seabird sources?

P7 L20: If the highest concentrations of salt aerosol are in the winter (as shown) then it makes sense that M&S OM is not correlated with sea salt. But, what about only in the summer months when M&S OM is elevated?

P7 L28: Do you mean that gas phase ammonia is neutralizing acidic particle phase species?

P7 L32: More discussion on what the "CHNO fragments" detected by Schmale et al. 2013 indicate is warranted here. How does it relate to your measurements?

P7 L32-33: More than simply the wind rose shown in the supplement is needed to make this statement. The authors can likely easily show that when these signals were high the wind was indeed bringing air from Cape Crozier.

P8 L3: It is not clear why a coastal source is suddenly implicated here. Is there evidence from back trajectories to show this? Is the proposed coastal source different from the penguin colonies?

P8 L13-14: It is not immediately clear that this is true.

P8 L26: What is meant by "emission concentrations?"

P8 L27: Is this the mean summer OM concentration?

P9 L11: The authors should provide a URL or, ideally, a DOI for this data.

P10-12: There are some issues with the reference formatting (e.g., missing DOI's, all caps titles). I believe that DOI's should be prefaced with "doi:" or shown as an active link (i.e., http://dx.doi.org/xxxx. . .). Please check through the references.

[Figure]

Table 1: Is a table the best way to display this information? Box and whisker plots would facilitate a better visual comparison of the data across seasons.

Figure 1: Can inter-quartile ranges be shown here?

Figure 5: Barrow marine and Alert marine spectra from $1500 - 1800$ cm$^{-1}$ more closely resemble the FFC spectrum rather than the M&S spectrum. How do the authors explain this? What does this imply about the assignment of the FFC source? Also, where does particle phase ammonium appear in this spectrum?

Figure 6: Does this correlation necessarily say that this fraction of the OM is driven by secondary processes? Could it be that the source of particle phase species coincides in time with available solar radiation? Do the peaks indicating N-containing species correlated with solar irradiance? Do any other organic functional group correlate with solar irradiance?

---

## Referee Comment (RC2) · Anonymous Referee #2 · 19 Feb 2018

The current paper reports organic concentration and speciation of aerosol organic functional groups during AWARE. There is a lack of measurements in the study region and these measurements are very timely. However, the paper is poorly written and it is not recommended for publication in ACP. A more specialized lower impact factor journal is recommended.

Major issues:

- The paper does not present any introduction of the aerosol composition in Antarctica. Mostly, the introduction seems a report of the history of a specific station. - Scientific questions and objectives are really not presented, neither discussed or summarized.

[Figure]

- After reading a paper a number of times, and looking at the figures, one can argue the main results are the impact of sea birds and marine sources in Antarctica. Not surprisingly at all, the reader does not understand if this is simply a bad measurement site (bad luck) or if the study has any implication.

I am afraid I cannot be any positive at this stage.

---

## Author Comment (AC1) · 20 Feb 2018

General Comments:
This manuscript provides an overview of one year of measurements of aerosol number concentrations, cloud condensation nuclei, and organic mass and functional groups (by FTIR) made near McMurdo Station at the southern end of Ross Island in Antarctica from November 2015 to December 2016. The authors characterize the sources of organic aerosol, and are able to establish some evidence for the contribution of both primary marine organic aerosol and the formation of secondary organic aerosol. Both the seasonal coverage of these measurements and the general lack of measurements in this region make these measurements very valuable and deserving of publication in ACP if the following issues can be addressed.

*We thank the referee for the well-considered comments that have been provided. We addressed the comments and concerns raised by the reviewer, and these have improved the manuscript substantially, as itemized below. **(Page and line numbers in this response reference the location in the discussion paper where the text is inserted. The revised manuscript will note these revisions separately with tracked changes when it is posted.)***

Major Comments:
1) This manuscript generally lacks clarity in the text and is difficult to follow. Many paragraphs contain a number of seemingly disparate thoughts or concepts. I suggest that the authors give this manuscript a very thorough revision with particular attention to the structure of the text.

*We thank the referee for the suggestion. We went through the text and revised several paragraphs for coherence and clarity. We emphasized key points more clearly, reorganized two sections, and clarified transitions to improve the flow in the text. Some of these changes also address the Minor Comments and are addressed in detail in the following pages; in addition, the following changes were made. Specifically, we have revised the following:*

*a) Clarified relevance of first paragraph of the Introduction in response to Minor Comments for P1 Line 30-31.*

*b) Improved transition at the end of the second paragraph in the Introduction in response to Minor Comments for P1 Line 30-31.*

*c) Improved transition at then end of the introduction in response to Major Comment 2 at P3 Line 9.*

*d) Improved transition of the first two paragraphs of section 3 in response to Major*

*Comment 3 and Minor Comments for P5 Line 1-22.*

*e) Improved organization of the fourth paragraph in section 3 in response to Minor Comments for P5 Line23-33.*

*f) Added summary of reasons for attribution of FTIR PMF factor to seabird and marine sources in response to Major Comment 5 at P8 Line 6.*

2) The scientific questions being addressed in this work need some clarification. I agree completely with the author's statement on P3L2-4 that this is most thorough yearlong aerosol data set from Antarctica. However, the questions that are addressed with this unique dataset are not clear, especially at the end of the introduction. After reaching the end of the manuscript it was more clear that the authors aimed to (1) characterize the sources of aerosol across seasons in Antarctica - especially of organic aerosol - and (2) establish evidence for the contribution of primary versus secondary formation of this organic aerosol. This could be stated more clearly at the end of the introduction.

*We agree that the end of the introduction could be more informative and have revised the paragraph as suggested: "This manuscript characterizes the sources of organic aerosol across four seasons in Antarctica and provides evidence for contributions to this organic aerosol from both primary sources and secondary processes." (P3 Line 9)*

3) Related to point (2), above, the authors spend a considerable amount of the text describing how the local pollution aerosol was removed from the data set. Once this aerosol signal has been removed (only from the total CN measurements, as far as I can tell) the authors then state (P5) that the remaining aerosol is from natural sources. It is very difficult to make this statement without first walking the reader through all the evidence you have for this point (e.g., how can you be sure that the "background" concentrations you measure are not impacted by local pollution). Further, the authors use a "despike" algorithm to remove high values of CN from the total particle con-centrations, but do not filter any of the other data (i.e., CCN concentrations). Do you obtain similar results if you use a wind-sector and wind-speed based filter to remove contamination from local pollution sources? How does a wind-sector filter impact CCN concentrations?

*We agree this question should be clarified so we added the following text for clarity: "There are two reasons why the CN concentrations that remain after SLCE (spikes) are removed are considered representative of the natural background rather than local pollution from McMurdo Station activities: First, the SLCE CN concentration is correlated weakly to BC absorption (r=0.48), but the background CN is correlated negatively to BC absorption (r=-0.4). Second, the two indicators of combustion-related pollution (BC absorption and the FFC factor) were approximately two times higher in summer than winter (Table 1), which is similar to the two-fold increase in SLCE CN in summer compared to winter but not enough to account for the seven-fold increase in the background (SLCE-removed) CN in summer compared to winter. Consequently, this larger summertime difference in background CN is likely associated with the higher*

*productivity of natural sources in summer. More specifically, the CN concentration associated with natural sources was very low (~60 cm$^{-3}$) in winter during low phytoplankton activity but as high as 2000 cm$^{-3}$ in summer (Figure S1), indicating a significant increase in biogenic (sulfate or organic) CN." (P4 Line 25)*

*"The spike-removal method was not applied to the CCN data because the CCN measurements did not have short-term spikes even at the highest supersaturation level (1%). The 1% SS CCN had only 0.1% of measurements that were 5% higher than the background (spike-removed) CN, supporting the assertion that particles from the local emissions were not active CCN in this study. The wind rose of both CCN concentration and fraction showed no directional preference. We added the following discussion to clarify this explanation: "The CCN measurements did not have short-term spikes even at the highest supersaturation level (1%), at which only 0.1% of the measurements were 5% higher than the background CN. The absence of the SLCE in the CCN measurements is likely the result of the local pollution being both too small and too low hygroscopicity to serve as CCN at 1% or below." (P5 Line 12)*

4) The Results and Discussion sections (Section 3 and 4) seem to provide only very cursory discussions of the results. These sections also contain some topics (e.g., PMF details including seeds and Fpeaks) that are better placed in the methods section. Further, the authors seem to assume a significant amount of prior knowledge about the methods and measurement location, all information needed to understand and interpret these measurements should appear in the paper or the supplement.

*We agree that we may have erred on the side of brevity in presenting these results and have added additional discussion on the topics specified in the minor comments as well as the topics below to the text. Specifically, we have revised the following:*

*a) Additional discussion of OM seasonality and comparison to Arctic: " Similar to CN concentrations, OM was highest in summer (0.27 µg m$^{-3}$) and lowest in winter (0.04 µg m$^{-3}$). Arctic OM at Barrow and Alert showed a very different seasonal pattern with low concentrations in Arctic summer (0.03 µg m$^{-3}$ and <0.5 µg m$^{-3}$ in Alert and Barrow, respectively) and high concentrations in winter and spring (0.3 µg m$^{-3}$ and 1 µg m$^{-3}$ in Alert and Barrow, respectively) (Frossard et al., 2011; Leaitch et al., 2017).  Consistent with OM, CN concentrations at these two Arctic sites, with particle size range of 80-500 nm at Alert and >100 nm at Barrow, were also low in Arctic summer (<50 cm$^{-3}$ and 100-300 cm$^{-3}$ at Alert and Barrow, respectively) and high in winter and spring (>100 cm$^{-3}$ and 400-1000 cm$^{-3}$ at Alert and Barrow, respectively) (Croft et al., 2016; Polissar et al., 1999).  The springtime high concentrations in the Arctic result from long-range transport from mid latitudes prior to the retreat of the Arctic front. The lack of substantial pollution sources at southern mid-latitudes (compared to those at northern mid-latitudes) means the Antarctic does not have an equivalent haze in spring (Stohl, 2006; Stohl and Sodemann, 2010; Russell and Shaw, 2015) . The higher summer OM in Antarctica is likely produced by the specific local conditions of the three polar sites, namely Ross Island has higher marine and seabird activity compared to Barrow and Alert." (P6 Line 10)*

*b) Improved explanation and comparison of marine-related FTIR PMF factors in response to Major Comment 5 at P8 Line 6.*

*c) Additional interpretation of the carboxylic acid group contribution and its seasonal and radiation dependence in response to Minor Comments for P8 Line 13*

*d) Clarification of the marine and FFC contributions to amine and ammonium in response to Minor Comments for P7 Line 22.*

*e) Discussion of the PMF analysis details was moved to the Method section based on Minor Comments at P5 Line 3 and P6 Line 13.*

*f) Discussion of the de-spike analysis details was expanded and moved to the Method section based on Minor Comments at P5 Line 3.*

*g) Additional aerosol sampling details were added in response to Minor Comments at P3 Line 20.*

*h) Clarification of OM calculation and detection limit for FTIR measurement in response to Minor Comments at P4 Line 3 and P7 Line 19.*

5) The attribution of the "Marine and Seabird" PMF factor is not very well discussed. I do not dispute the claim, as it does appear likely given the proximity of large penguin colonies. Something as simple as a wind direction analysis relating high concentrations of the M&S factor with winds from the direction of penguin colonies would help your argument. Providing an analysis of back trajectories is also advisable.

*We have revised the text to state more clearly the reasons why marine and seabirds are the most likely the sources of the M&S OM as follows: "There are four reasons that the M&S factor are likely associated with marine and seabird emissions: The 1680 cm$^{-1}$ signal has been found at two coastal Arctic sites (in small amounts) but not on open ocean marine studies (Leaitch et al., 2017; Shaw et al., 2010; Frossard et al., 2011; Russell and Shaw, 2015). This difference suggests that the amide group is likely associated with seabirds, since they are found in coastal marine areas but generally not in open ocean marine areas. The 1680 cm$^{-1}$ signal has been found at two coastal Arctic sites (in small amounts) but not on open ocean marine studies (Hawkins and Russell, 2010; Leaitch et al., 2017; Shaw et al., 2010; Frossard et al., 2011). This difference suggests that the amide group is likely associated with seabirds, since they are found in coastal marine areas but generally not in open ocean marine areas. The higher concentrations of the M&S OM factor coincided with the summer breeding period of a large penguin colony at Cape Crozier, which was upwind during most of the summer. Other possible contributions, such as from algal blooms during ice melting in spring, are not consistent with the northeasterly winds, the amide group, or the seasonality of the M&S OM." (P8 Line 6)*

*We thank the referee for the suggestion to look at the wind pattern and back trajectories. We did carry out both wind direction and back-trajectory (HYSPLIT) analysis for this study. The HYSPLIT results did not add useful information mostly because of the daily variability of the trajectories and the weekly time resolution of the filters. The wind direction results are also limited by the time resolution of the filters, such that the weekly average wind direction was always essentially northeasterly. Specifically, while the M&S fraction of OM is higher when wind is from northeasterly sectors, the FFC showed a similar pattern (see Figure below). This is consistent with our assertion that much of the FFC could be associated with local road traffic, which was not associated with a particular wind direction. This is clarified as follows: "HYSPLIT back trajectories (Draxier and Hess, 1998) did not add useful information because the day-to-day variability exceeded the differences among weekly averages. Weekly-average wind direction was always northeasterly (±45 degrees), so there was insufficient variation to identify sources in different directions. " (P7 Line 20)*

[Figure]

Minor Comments:
P1 L30-31: This line reads as though the authors are implicating aerosol to some extent in the melting of the Antarctic ice sheet. I do not dispute that there can be a connection between atmospheric composition and ice dynamics (e.g., through cloud processes), but these processes are not nearly as well understood as the authors seem to suggest here.

*We agree with the referee and revised the paragraph as follows: " West Antarctica is one of the most rapidly warming regions on Earth (Bromwich et al., 2013) , which has potential impacts for the melting of the Antarctic ice sheets and consequent sea level rise (Steig et al., 2009; Lambeck et al., 2002) . In some regions, ambient aerosols contribute substantially to the radiation balance (Stocker et al., 2013) , but little is known about the sign and magnitude of their contribution in Antarctica because of the lack of measurements of their abundance, composition, and sources. In fact, there are few places on Earth where measurements of aerosols and their properties are needed to constrain*

*modeled radiation as much as in Antarctica." (P1 Lines 28-31)*

P2 L1-21: This paragraph reads as a list of what has been done by others in this region. I agree this such a summary is an important part of this manuscript, but the introduction generally lacks a take-away message. For example, do the authors believe that we can say from past measurements that regional marine biological emissions are an important contributor to Antarctic aerosol? Do we know more about the biological sources of sulfate than organic aerosol? How does this help to motivate your study?

*We clarified this by adding the following discussion: "Biological sulfate aerosol accounts for 43–65% of the summer zonal mean CCN concentrations and 7–20% of the winter CCN over the oceans in the Southern Hemisphere, including the circumpolar Southern Ocean (Korhonen et al., 2008). This important role for biological sulfate in the Southern Ocean suggests that biogenic organic components may also contribute significantly to particle number and mass, but measurements of organic particles are too scarce to determine if this is the case (McCoy et al., 2015). " (P2 Line 15)*

P2 L27 (and Table S1): What do we learn from this collection of amino acid measurements?

*We revised the description of Table S1 to better explain its relevance to this study as follows: "Some of the few measurements of organic aerosol particle composition that have been made in marine and polar regions are those of amino acids, which are summarized in Table S1 (Mace et al., 2003a; Kuznetsova et al., 2005; Scalabrin et al., 2012; Barbaro et al., 2015; Mace et al., 2003b; Wedyan and Preston, 2008; Shi et al., 2010; Matsumoto and Uematsu, 2005; Mandalakis et al., 2011; Violaki et al., 2010) Amino acids in remote marine and coastal regions have been used as markers for biological activities since they are natural chemical constituents of many marine and terrestrial organisms (Barbaro et al., 2015; Scalabrin et al., 2012; Milne and Zika, 1993; Cowie and Hedges, 1992). In addition, amino acids contain organic nitrogen and specifically amine groups, which are also consistent with measurements in polar regions of CHNO fragments (Schmale et al., 2013) and amine groups (Shaw et al., 2010; Frossard et al., 2011) "*

P3 L 11-20: Rather than referring to a web link for further information on the sampling site, and pertinent information should be summarized here. Information about inlets? Length, flow rates, transmission efficiency as a function of size?

*We have added the following text to describe the aerosol-related measurement details: "The aerosol inlet samples at ~10 m above ground level and has a rain guard and bug screen, 1000 L min$^{-1}$ turbulent flow through 4.6 m of large-diameter (20 cm ID), powder-coated aluminum tubing, a 2.1 m smaller-diameter tube (4.76 cm ID) that extracts 150 L min$^{-1}$ flow from the center of the larger-diameter tubing, and a flow distributor with five ports, each drawing 30 L min$^{-1}$ through 25 cm of 1.59 cm (5/8") inner diameter stainless-steel tubing. The size-dependent losses were measured below 10% for particles from 10*

*nm to 10 um diameter (https://www.arm.gov/publications/tech_reports/doe-sc-arm-tr-191.pdf)." (P3 Line 20)*

P4 L3: The detection limit needs to be defined in order for the statement "above the detection limit" to convey any information.

*We have added this definition as "The detection limit and error for each functional group is the larger of twice the standard deviation of the absorption values associated with blank filters and the visual determination of the minimum peak size that could be distinguished from spectral noise (Maria et al., 2002). The detection limit of OM was 0.09 μg based on the sum of the detection limits of the three largest functional groups during the project (alkane, hydroxyl and amine). For the weekly air sampling volume of 80 m$^3$ used in this study, this loading corresponds to a concentration of 0.001 μg m$^{-3}$." (P4 Line 3)*

P4 L29: CN has not been defined at this point - I assume this is the CPC measurement?

*We have added this definition as "CN (condensation nuclei from CPC)" (P4 Line 29)*

P5 L1-22: The topic of this paragraph is not clear. There is a mid-paragraph shift from discussing pollution aerosol and its removal from the data set to discussing "background" aerosol.

*We agree this section needed to be re-organized and revised; we have done this as follows: As suggested in the next comment (P5 L3), we moved part of this paragraph to the Method to distinguish it from the Results. The Results part is now organized in two paragraphs, the first describes why the SLCE are considered pollution and the second explains why the background CN are likely largely from natural sources. The revised text at the beginning of section 3 now reads: "19% of the 1-Hz CN measurements recorded during the project were identified as SLCE, and the average of the concentrations for those times contributed 55% of the project-average CN concentrations. The distribution of SLCE duration and timing (Figure S7) shows that SLCE events were approximately two times more frequent during local daytime than nighttime. This short duration and largely daytime timing of SLCE suggests that site maintenance and nearby road traffic are likely responsible for many of the high CN events." (P4 Line 25)*

*"There are two reasons why the CN concentrations that remain after SLCE (spikes) are removed are considered representative of the natural background rather than local pollution from McMurdo Station activities: First, the SLCE CN concentration is correlated weakly to BC absorption (r=0.48), but the background CN is correlated negatively to BC absorption (r=-0.4). Second, the two indicators of combustion-related pollution (BC absorption and the FFC factor) were approximately two times higher in summer than winter (Table 1), which is similar to the two-fold increase in SLCE CN in summer compared to winter but not enough to account for the seven-fold increase in the background (SLCE-removed) CN in summer compared to winter. Consequently, this larger summertime difference in background CN is likely associated with the higher*

*productivity of natural sources in summer. More specifically, the CN concentration associated with natural sources was very low (~60 cm$^{-3}$) in winter during low phytoplankton activity but as high as 2000 cm$^{-3}$ in summer (Figure S1), indicating a significant increase in biogenic (sulfate or organic) CN. " (P4 Line 25)*

P5 L3: This "despike" algorithm need to be described in more detail in the methods section. Since the goal of this paper is likely not to demonstrate the effectiveness of this "despike" algorithm, this isn't really a result but rather a part of data quality control that is applied before any interpretation can be made.

*We moved this discussion to the Methods section and added further details as follows: " CN (condensation nuclei from CPC) concentrations had frequent short-lived increases that typically had high concentrations (>1000 particles cm$^{-3}$ for 1 Hz CN), which we attributed to short-term local contamination events (SLCE) (Figure S1). High CN concentrations (>1000 cm$^{-3}$) occurred 48% of the time when the wind was from the west (Figure S2), which is the same direction as the McMurdo Station central facilities. However, westerly winds only occurred 3% of the time, so emissions at McMurdo Station were unlikely to account for most of the emissions. Spikes were separated using a "de-spike" algorithm based on running median filters (Beaton and Tukey, 1974; Tukey, 1977; Velleman, 1977; Goring and Nikora, 2002) . We applied a running median length of 24 hr and weighted by cosine bell running mean of 24 hr to the 1 Hz CN concentration and assigned the CN concentration above the resulting filter as SLCE. The SLCE were characterized by an average duration of less than 1 hr (0.5 min$\pm$6 min), rapid rate of concentration change (8520$\pm$36780 cm$^{-3}$ min$^{-1}$), and concentrations exceeding 1000 cm$^{-3}$. After SLCE (spikes) were removed, the 24-hr running median concentration was interpreted to be the natural background CN, for reasons discussed in Section 3. " (P3 Line20)*

P5 L5: SLCE needs to be defined.

*We have added this definition as "short-term local contamination events (SLCE)" in the Method section. (P3 Line 20)*

P5 L5-6: What does "accounted for" mean in this context? You need to provide a more quantitative assessment to make this statement. Further it is unclear that is meant by "occurred 19% of the time."

*We have revised this text as follows: "19% of the 1-Hz CN measurements recorded during the project were identified as SLCE, and the average of the concentrations for those times contributed 55% of the project-average CN concentrations. " (P5 Line 5)*

P5 L7: A mean ± standard deviation doesn't really tell you that the change was rapid in time, a rate of change might be a better metric.

*We have clarified that these numbers are the rate of change with units of cm$^{-3}$ min$^{-1}$ as follows "...rapid rate of concentration change..." (P5 Line 7)*

P5 L11: Is this the author's hypothesis? If so, it should be stated as such. At this point in the analysis, it seems difficult to make this statement. For example, the remaining CN could be from background pollution not immediately associated with plumes.

*We interpret this comment to be about the reasons for identifying non-SLCE CN as natural. We clarified this point by the following text: "There are two reasons why the CN concentrations that remain after SLCE (spikes) are removed are considered representative of the natural background rather than local pollution from McMurdo Station activities: First, the SLCE CN concentration is correlated weakly to BC absorption (r=0.48), but the background CN is correlated negatively to BC absorption (r=-0.4). Second, the two indicators of combustion-related pollution (BC absorption and the FFC factor) were approximately two times higher in summer than winter (Table 1), which is similar to the two-fold increase in SLCE CN in summer compared to winter but not enough to account for the seven-fold increase in the background (SLCE-removed) CN in summer compared to winter. Consequently, this larger summertime difference in background CN is likely associated with the higher productivity of natural sources in summer. More specifically, the CN concentration associated with natural sources was very low (~60 $cm^{-3}$) in winter during low phytoplankton activity but as high as 2000 $cm^{-3}$ in summer (Figure S1), indicating a significant increase in biogenic (sulfate or organic) CN." (P4 Line 25)*

P5 L13: "...correlated with..."

*We think the reviewer is suggesting that we use the phrasing "Correlated with" rather than "correlated to". Both are considered standard English (per Merriam-Webster), so we kept "to" as this phrasing suggests a more specific quantitative relationship. (P5 Line 13) In the event we misunderstood this comment, we welcome clarification.*

P5 L19 (Figure S1): Actually plotting CCN/CN would be informative, rather than asking the reader to estimate this ratio by looking at separate plots of CCN and CN

*We added a panel showing CCN/CN as suggested in Figure S1.*

[Figure]

*Figure S1. Concentrations of: (a) measured CN, (b) SLCE-removed CN and measured CCN, and (c) ratio of CCN to SLCE-removed CN.*

P5 L19-22: The meaning in these two sentences is difficult to discern. My sense is that the authors are trying to make a statement about changing CCN activation diameters, but it is unclear (i.e., what "decrease in particle diameter"? Associated with what phenomena?). Could the change in CCN/CN ratio result from either a change in particle diameter in summer or a change in composition that results in a change in the activation diameter?

*We have revised this discussion to clarify as follows: "..but from late September to early October the ratio of CCN/CN decreased to 0.5 at 1% supersaturation (Figure S1). This decrease of the ratio of CCN to background (spike-removed) CN during the winter-spring transition could be caused by changes in particle size and composition. One such cause would be additional CN that are too small to contribute to CCN. Previous observations at a site 10 km from McMurdo Station showed an increase in the fraction of CN smaller than 250 nm at polar sunrise (September-October), although a specific cause was not clear (Giordano et al., 2017). The higher CCN/CN ratio in the summer (Table 1) is consistent with both the higher biogenic sulfate contributions during the highest productivity season (summer) and the slightly larger diameter of the accumulation mode particles observed in previous summers (Kim et al., 2017)." (P5 Line 19-22)*

P5 L23-33: This paragraph begins by discussing aerosol growth factors and ends by discussing aerosol absorption. The connection the authors are trying to make, if any, in unclear.

*The connection is that BC is negatively correlated with activation of CCN. We have revised the following sentences here to clarify this point: " The particles that had too low hygroscopicity to grow measurably may be those that were emitted by local anthropogenic emissions. The moderate correlation of BC absorption to the fraction of particles that did not grow at increased relative humidity in the HTDMA (R=0.52, Figure 2 (a)) indicates that the BC-containing particles could be the particles that have low hygroscopicity. In addition, BC absorption correlated moderately to the non-activated CN particles (1-CCN/CN) (R=0.34 for 1% supersaturation, Figure 2 (b)). Since BC-containing particles, such as those freshly emitted from combustion sources, have been shown to have low hygroscopicity (Peng et al., 2017; Vu et al., 2017), these correlations are consistent with the particles that did not take up water being those that were emitted by local combustion activities." (P5 Line 26-33)*

P5 27-29: The meaning of this sentence is unclear (i.e., a much smaller change in what?).

*We have revised this discussion to clarify as follows: "Second, the two indicators of combustion-related pollution (BC absorption and the FFC factor) were approximately two times higher in summer than winter (Table 1), which is similar to the two-fold increase in SLCE CN in summer compared to winter but not enough to account for the seven-fold increase in the background (SLCE-removed) CN in summer compared to winter."(P5 Line 15)*

P6 L2: Figure 2 is referred to in the text before Figure 1.

*We agree this is true on P6, but Figure 1 was referred to first on P4, so we believe the current numbering is correct.*

P6 L3-6: Is the sea salt measured at this cite during winter transported form open water areas? What evidence do the authors have for this? The authors go on to suggest that frost flowers are the major source of sea salt at this site, but more discussion is warranted here. While, sodium to chloride or sodium to sulfate ratios provide some evidence, I do not see that this rules out other sources of aerosol, such as blowing snow, which has been observed in Antartica (e.g., Jones et al., 2009 www.atmos- chem-phys.net/9/4639/2009/). Field observations and laboratory experiments seem to suggest that frost flowers are rigid and difficult to break, even at wind speeds up to 12 m/s (e.g., Yang et al., 2017 https://doi.org/10.5194/acp-17-6291-2017; Roscoe et al., 2011 doi:10.1029/2010JD015144). A recent model study suggests that the frost flower source of sea salt aerosol cannot explain the seasonality of sea salt aerosol across several Arctic stations (Huang et al., 2017 doi:10.5194/acp-17-3699-2017). A more thorough discussion of possible sources supported by meteorological variables and back trajectories is needed to draw a conclusion here, especially since upward migration of brine and incorporation

of frost flowers can lead to depletion of the sulfate-to-sodium ratio relative to bulk sea water such that this chemical signature may not be unique to frost flowers.

*We have investigated the source question with both wind direction at the site and back trajectory calculations. Both analyses were limited by the one-week sampling time of the filters: The weekly-averaged wind direction was consistently northeasterly so there was no variability to associate with different sources. The 1-day back trajectories were inconsistent, with some circular and others northeasterly, so that an average over a week was not meaningful. Therefore, we were unable to distinguish open-ocean, sea ice, or land sources based on meteorology. This is clarified as follows "HYSPLIT back trajectories (Draxier and Hess, 1998) did not add useful information because the day-to-day variability exceeded the differences among weekly averages. Weekly-average wind direction was always northeasterly ($\pm$45 degrees), so there was insufficient variation to identify sources in different directions." (P7 Line 20)*

*Other indicators of sources, i.e. particle composition and wind speed, were also considered here. We have added the following discussion to clarify this analysis: " The measured $Cl^-/Na^+$ of 2 represents a large sodium deficiency in wintertime submicron particles (Figure 1). The depletion of $Na^+$ relative to $Cl^-$ in winter indicates a likely*

*contribution to the aerosol submicron mass from wind-blown frost flowers (Alvarez-Aviles et al., 2008; Thomas and Dieckmann, 2003; Stein and MacDonald, 2004; Papadimitriou et al., 2007; Giannelli et al., 2001; Belzile et al., 2002; Shaw et al., 2010). This sodium depletion is the result of $Na_2SO_4$ precipitating out from sea ice brine before frost flowers wick up the remaining salt solution. Blowing snow could also contribute to submicron particles (Domine et al., 2004) , but this source has not been associated with a substantial sodium deficiency in submicron particle composition (Gordon and Taylor, 2009)" (P6 Lines 6)*
*"If either frost flowers or blowing snow were generated near the site, we would expect a correlation of concentrations to wind speed at higher wind speeds, since both sources have been characterized as requiring wind speed thresholds of approximately 7 m s$^{-1}$ for lofting of particles (Schmidt, 1981; Shaw et al., 2010) . During AWARE, 1-min wind speed only exceeded this threshold by 1 m s$^{-1}$ for 24% of the time, and the weekly average wind speed was never higher than 7 m s$^{-1}$. Wind speed had no correlation to CN concentration for the campaign (r=-0.32) or for winter (r=-0.31). In addition, there was no correlation (R=-0.15) of submicron CN number with wind speed (>8m s$^{-1}$), as would be expected for blowing snow generated locally (Yang et al., 2008) . The M&S factor concentration also showed no correlation (r=0.1) to the fraction of time with high wind speed (>8 m s$^{-1}$). While these relationships do not support the attribution of the wintertime salt mass to either frost flowers or blowing snow, they do not rule it out since the particles may have been lofted upwind and transported to McMurdo Station." (P6 Lines 6)*
*" A recent model simulation (Huang and Jaegle, 2017) predicted that blowing snow has significantly higher contributions to submicron particle mass than frost flowers in Antarctica and the Arctic, but also showed that the region at the north edge of the Ross Ice Shelf (including Ross Island) had both higher emissions (>0.6 10$^{-6}$ kg m$^{-2}$ d$^{-1}$) and concentration (>1.5 μg m$^{-3}$) from frost flowers than the emissions (<0.4 10$^{-6}$ kg m$^{-2}$ d$^{-1}$)*

*and concentration (<1.0 µg m$^{-3}$) from blowing snow, consistent with the finding that wintertime OM at McMurdo Station were more likely from frost flowers than blowing snow." (P6 Lines 6)*

P6 L11: Define what "baselined" means in this particular case. This should probably be discussed in the methods section.

*We have improved the description of baselining as follows: "FTIR spectra were baselined by subtracting a combination of piecewise linear and polynomial regressions from the spectrum using an automated algorithm (Takahama et al., 2013) ." (P6  Line11)*

P6 L13-15: These specific belong in the methods section. In the Results and Discussion section I suggest you discuss why your choice of two factors is the physically most meaningful choice (some of this discussion exists, but could be expanded).

*As suggested we moved the following discussion to the Methods section: "Positive Matrix Factorization (PMF) was applied to the baselined FTIR spectra for the PM$_1$ samples collected in 2016 at McMurdo Station with the PMF2 V4.2 (Paatero and Tapper, 1994; Paatero, 1997). Six-factor solution spaces (1~6) were considered. Fpeak values from -2 to 2 at 0.5 increments were considered. Seeds of 1, 10 and 100 were used at each Fpeak and factor number to examine the robustness of each solution. There was little change in solutions with rotations for all solutions. Q/Qexpected decreases as factor number increases for all solutions (Table S2). The two-factor solution is considered robust because the spectra are almost identical for all rotations and seeding conditions (Figure S6). The solution leaves an average of 23% of the OM as residual. The two factors are not correlated in time and do not have similar spectra (Table S2). The new factor identified from the 3-factor solutions is either degenerate or very similar (cosine similarity =0.99) to one of the first two factors. Similarly, for 4 or more factors, solutions contain two or more degenerate or duplicate factors. This makes the two-factor solution with Fpeak of 0 optimal for the AWARE data set. The low aerosol concentrations and limited personnel access at AWARE reduced the time resolution of FTIR samples to one week each for a total of 54 samples in one year, which means the resolution of the factor separation is significantly lower than has been possible in other regions (Russell et al., 2011); the small number of samples and the low variability during the study is likely the reason that PMF is unable to separate more than two factors." (P3 Line 30)*

*"In addition, K-means clustering (Hartigan and Wong, 1979) was applied to the baselined FTIR spectra (Takahama et al., 2013). Solutions with 1 to 10 clusters were evaluated. The 2-cluster solution was chosen because solutions with 3 or more clusters included at least one pair of clusters with centroids with cosine similarity higher than 0.95 (Table S2), making those clusters effectively overlapping."  (P3 Line 30)*

P6 L 24-25: The motivation for carrying out a PMF analysis and a k-means cluster analysis of the FTIR spectra is not adequately explained here. Presumably if these two approaches yield similar results, then this lends confidence to assignment of source types. This information needs to be stated clearly.

*The reviewer is correct and so we have added the following discussion to explain the implications of the factor-cluster comparison: " Factorization techniques like PMF are applied to separate each individual composition measurement into the independent factors that contribute to its composition, where these factors may represent different sources as well as different formation processes. On the other hand, clustering algorithms are used to sort similar measurements into categories, each of which may contain a mixture of different sources and formation processes and is characterized by the centroid to which all measurements in that category are most similar. The similarity of the k-means centroids and PMF factors (cosine similarity > 0.97) indicates that both separations are robust. " (P3 Line 30)*

P6 L25-26: This needs to be actually shown in some way, even in the supplement.

*We revised Table S2 and the supplemental discussion as follows: "The 2-cluster solution was chosen because solutions with 3 or more clusters included at least one pair of clusters with centroids with cosine similarity higher than 0.95 (Table S2), making those clusters effectively overlapping" (P3 Line 30)*

**Table S2. Parameters for FTIR PMF factor and K-means clustering evaluation.**

| Number of Factors / Criteria | 2 | 3 | 4 | 5 | 6 |
|---|---|---|---|---|---|
| $Q/Q_{exp}$ | 7.06 | 6.02 | 4.75 | 3.90 | 3.25 |
| Absolute residual | 23.6% | 21.7% | 17.4% | 14.2% | 12.0% |
| Temporal correlation factor strength (r>0.8) | None | None | None | None | None |
| Number of similar factor spectra (Cosine similarity>0.8) | None | 1 pair | 1 pair | 2 pairs | 4 pairs |
| Factors with less than 10% OM | None | None | None | 1 | 1 |
| Number of similar cluster centroids (Cosine similarity>0.95) | None | 1 pair | 3 pairs | 4 pairs | 6 pairs |

P6 L32 (Table 1): In addition to stating the observation that the residual is much larger than the MS factor in winter, the authors should discuss what is means for interpretation of their results. This comment applies to all "ratios" provided in Table 1, the authors state that this provides a measure of uncertainty, how do the authors want the reader to interpret this? For example, does this result suggest that the PMF factors are robust across seasons? Only in specific seasons? This information should be clear.

*As the reviewer has suggested, the uncertainty of the PMF separation to different factors is different by season, which we have clarified as follows: "Since the PMF residual is the fraction of OM that could not be assigned to either factor, the ratio of the residual to the*

*factor OM provides a measure of the uncertainty of the PMF separation – namely the fraction of OM that could be missing from the factor. The ratio of the PMF residual to the FFC OM varies from 29% in winter to 63% in summer, making this result more likely to represent all of the FFC OM in winter when FFC OM is a larger relative fraction of OM. Similarly, the PMF residual is 33% of M&S OM in summer, indicating the source separation could be missing a third of M&S OM.  In contrast, the PMF residual is 9 times larger than the M&S OM in winter (Table 1), making the quantification of M&S OM in winter very uncertain." (P6 Line 32)*

P7 L11: Is the FTIR method sensitive to inorganic ammonium? If so, can a spectrum of e.g., ammonium sulfate be included to support this hypothesis.

*To address this question, we have added this text as follows: "The ammonium mass is not quantified by FTIR of Teflon filter samples because ammonium nitrate is semi-volatile. The location of absorption by sulfate in FTIR coincides with the location of Teflon absorption. Since the absorption by the Teflon filter far exceeds that of the sulfate particles, sulfate cannot be measured on this substrate. Sulfur was measured by XRF and is expected to be largely ammonium sulfate, since organosulfate and bisulfate were below the limit of quantification." (P3 Line 30)*

P7 L15: "80 % hydroxyl group" - Is this a fraction of the total organic mass? How do you know that all organic mass is accounted for? Do you have to assume an average molecular weight for aerosol components? More description of how you arrive at these values is warranted (in the methods section).

*"OM is calculated as the sum of all functional groups measured above detection, based on the assumptions of Russell (2003) . Subsequent evaluations and intercomparisons (Takahama et al., 2013; Russell et al., 2009; Maria et al., 2002) have shown that errors associated with functional groups that are not quantified because of Teflon interference and semivolatile properties are accounted for within the stated $\pm20\%$ uncertainty for ambient particle compositions." (P7 Line 15)*

P7 L15-16: Are these differences between Arctic and Antarctic summer aerosol significant? Do you expect a difference? More discussion is needed here, rather than just stating that there is a difference between these two, arguably rather different, regions.

*In addition to the comparison of the hydroxyl fractions and cosine similarities that is on P7 Line 12-14, we have added the following discussion: "Barrow and Alert had higher marine OM concentrations in winter than in summer. Likely this is because these two Arctic sites did not have the large seabird contributions that contributed to the M&S factor on Ross Island during summer (Lyver et al., 2014) .  The smaller seabird populations near the Arctic sites also meant that Barrow and Alert OM had only very small amide contributions (Figure 5)." (P7 Line 14)*

P7 L19: What is you detection limit for total organic mass?

*We have clarified the detection limit as follows: "The detection limit and error for each functional group is the larger of twice the standard deviation of the absorption values associated with blank filters and the visual determination of the minimum peak size that could be distinguished from spectral noise (Maria et al., 2002) . The detection limit of OM was 0.09 μg based on the sum of the detection limits of the three largest functional groups during the project (alkane, hydroxyl and amine). For the weekly air sampling volume of 80 m$^3$ used in this study, this loading corresponds to a concentration of 0.001 μg m$^{-3}$." (P4 Line 3)*

P7 L19: Is it not the other way around - high summer OM and low winter OM attributed to marine and seabird sources?

*Yes, high summer and low winter is correct. Thanks for pointing out. We have corrected this sentence as follows "The low winter and high summer M&S OM means…" (P7 Line 19)*

P7 L20: If the highest concentrations of salt aerosol are in the winter (as shown) then it makes sense that M&S OM is not correlated with sea salt. But, what about only in the summer months when M&S OM is elevated?

*We have clarified that: "Marine OM contributions could be high in winter relative to summer because of the higher regional wind speeds, but their absolute concentration was too low to separate and identify in this set of 54 one-week samples. Specifically, the small number of long-duration samples resulted in PMF residuals that were more than 9 times higher than the M&S factor in winter, so that the marine fraction in winter is very uncertain." (P7 Line 20)*

P7 L28: Do you mean that gas phase ammonia is neutralizing acidic particle phase species?

*Yes, that is correct, but we have re-phrased to emphasize uptake since we do not know if the resulting particles are neutral pH: "..., which is taken up on particles as ammonium." (P7 Line 28)*

P7 L32: More discussion on what the "CHNO fragments" detected by Schmale et al. 2013 indicate is warranted here. How does it relate to your measurements?

*We thank the reviewer for noting this and have revised the following discussion: "Previous studies have also attributed aerosol emissions and properties to penguin activities, including ammonia-enhanced new particle formation (Weber et al., 1998) and oxalate-enriched particles and organonitrogen-containing fragments from urea breakdown products (Legrand et al., 2012; Schmale et al., 2013). The finding here of amide groups would be consistent both with particle formation and with substantial organonitrogen components." (P2 Line 25)*

*We also note that the introduction has been revised to include: "CHNO fragments identified by mass spectrometry have been associated with uric acid and other nitrogen containing components that are produced from penguin guano (Schmale et al., 2013) " (P2 Line 33)*

P7 L32-33: More than simply the wind rose shown in the supplement is needed to make this statement. The authors can likely easily show that when these signals were high the wind was indeed bringing air from Cape Crozier.

*As noted above (Major Comment 5), wind came from Cape Crozier during most of the summer: "The higher concentrations of the M&S OM factor coincided with the summer breeding period of a large penguin colony at Cape Crozier, which was upwind during most of the summer." (P7 Line 20)*

P8 L3: It is not clear why a coastal source is suddenly implicated here. Is there evidence from back trajectories to show this? Is the proposed coastal source different from the penguin colonies?

*We apologize for the confusion. We used coastal here because as noted above the amide signal has been found at two coastal Arctic sites (in small amounts) but not in open ocean marine studies. We have revised the text to read: "The 1680 cm$^{-1}$ signal has been found at two coastal Arctic sites (in small amounts) but not on open ocean marine studies (Hawkins and Russell, 2010; Leaitch et al., 2017; Shaw et al., 2010; Frossard et al., 2011) . This difference suggests that the amide group is likely associated with seabirds, since they are found in coastal marine areas but generally not in open ocean marine areas." (P8 Line 6)*

P8 L13-14: It is not immediately clear that this is true.

*We have revised this text as follows: "This difference may be because the local emissions from McMurdo Station facilities reached the Cosray site in less than 5 min (since McMurdo Station was 2 km away and wind speeds were 6 m s$^{-1}$ on average) making them essentially "fresh" primary particles, whereas those from the large upwind penguin colony took 6 hr (since Cape Crozier was 100 km away and wind speeds were 6 m s$^{-1}$ on average) to reach the site giving them approximately 50 times more time for photochemical reactions leading to SOA production. It is also possible that the anthropogenic gas-phase precursor emissions had lower SOA acid yields but there is little evidence to support this (Rickard et al., 2010; Wyche et al., 2009; McNeill, 2015) ." (P8 Line 13)*

P8 L26: What is meant by "emission concentrations?"

*We corrected this to read "concentrations". (P8 Line26)*

P8 L27: Is this the mean summer OM concentration?

*Yes, we clarified this by revising the text to read: "The mean summer OM concentration was..." (P8 Line 27)*

P9 L11: The authors should provide a URL or, ideally, a DOI for this data.

*As suggested, we have added the DOI as: https://doi.org/10.6075/J0WM1BKV (P9 Line 11)*

P10-12: There are some issues with the reference formatting (e.g., missing DOI's, all caps titles). I believe that DOI's should be prefaced with "doi:" or shown as an active link (i.e., http://dx.doi.org/xxxx. . .). Please check through the references.

*The referee is correct. We have checked and revised all references as suggested. "doi:" are added and caption titles are fixed.*

Table 1: Is a table the best way to display this information? Box and whisker plots would facilitate a better visual comparison of the data across seasons.

*The key variables in Table 1 are included for OM and factors in Figure 1 and for CCN and CN in Figure S1. The point of this table is to also include the specific average values and variability, which includes a large variety of numbers, which would likely also be quite cumbersome as a figure. For these reasons, we have retained the table.*

Figure 1: Can inter-quartile ranges be shown here?

*To show the variability as suggested, we added standard deviations in Figure 1 as below:*

[Figure]

*Figure 1. Monthly average of (a) Temperature, shortwave downwelling irradiance measured in this study and sea ice expansion rate of the Ross Sea (Holland, 2014) ; (b) Sea salt, dust and non-sea salt sulfate concentration from XRF and FTIR peak location at 1500~1800 cm⁻¹ wavenumber region. Standard deviations are shown on the plot as error bars.*

Figure 5: Barrow marine and Alert marine spectra from 1500 – 1800 cm−1 more closely resemble the FFC spectrum rather than the M&S spectrum. How do the authors explain this? What does this imply about the assignment of the FFC source? Also, where does particle phase ammonium appear in this spectrum?

*We have revised the labels on Figure 5 and have added the following discussion to clarify how these factor spectra are distinguished: "The primary amine peak (1620 cm⁻¹) is present in both FFC and M&S factors at McMurdo Station (Figure 5), consistent with previous studies (Shaw et al., 2010; Guzman-Morales et al., 2014; Price et al., 2017; Leaitch et al., 2017) . The difference between the FFC and M&S spectra is that FFC has double sharp alkane group peaks at 3000 cm⁻¹ but M&S has a broad hydroxyl group absorption at 3400 cm⁻¹ (Figure 4). Ammonium has peaks at 3050 and 3200 cm⁻¹ and contributes to both FFC and M&S spectra (Figure 4). " (P7 Line 9)*
*Figure 5:*

[Figure]

Figure 6: Does this correlation necessarily say that this fraction of the OM is driven by secondary processes? Could it be that the source of particle phase species coincides in time with available solar radiation? Do the peaks indicating N-containing species correlated with solar irradiance? Do any other organic functional group correlate with solar irradiance?

*The reviewer has identified an important point, and we have rephrased the text to better explain that the acid group is used as the indicator of secondary rather than primary OM since the radiation correlation could be explained either by secondary processes or by the seasonal dependence of the seabird emissions: "
[revised manuscript text omitted]

---

## Referee Comment (RC3) · Anonymous Referee #3 · 7 Apr 2018

Comments on Liu 2018 Antarctica aerosol

General Comments

This paper covers a year's worth of organic aerosol measurements in a region that is rarely sampled and sheds valuable insight into the chemical composition of Antarctic aerosol. The paper is well-written and the figures and tables are clear and legible. Aside from minor corrections and qualifications, I have just one concern with the current form of the manuscript. The authors claim a connection between carboxylic acid variability and downwelling radiation that, for reasons I describe below, is misleading. Unless the authors can clarify and justify this correlation, I would recommend that the

discussion of that connection be omitted. Otherwise the paper needs only minor revision before it is suitable for publication. Given that this aspect may take more work to revise, I am selecting major revision in the online evaluation. However, the changes should not be overly burdensome.

Specific Comments

Pg 2 line 22: please give a value range for the "high fraction" of OH observed previously

Pg 2 line 25: similarly, what range of ON mass fractions have been observed? Is it a minor component, or major? Or highly variable?

Pg 5 line 5: I don't see SLCE defined before it is use as an abbreviation

Pg 7 line 4: I'm unclear as to what the "factors identified as urban combustion emissions" is that correlates to the FFC factor. Do the authors mean "factor spectra"? Like, other FTIR PMF spectra? Please clarify.

Figure 6 and discussion on page 8 (and in conclusion/abstract): Drawing any relationship about photochemistry from the correlation between M&S carboxylic acid and DWR is misleading. The observed correlation, as I understand it, is simply the correlation between the M&S factor strength and down-welling radiation time series, since the carboxylic acid attributed to the M&S factor is always the same fraction of the factor (given in Fig 4), and so varies only as the strength of that factor. The same correlation coefficient (r) would be obtained for any of the functional groups present in the M&S factor and for the factor as a whole, as correlation coefficients do not change with addition/subtraction or multiplication by constants to the vectors being compare. Further, the downwelling radiation is varying only because of the season change (Fig 1) and the strength of the M&S factor, associated with the Adelie penguins, is also due to seasonal migration, so the observed correlation to downwelling is really just a product of the M&S factor and downwelling both having season characteristics. The authors would have to do more analysis and include other metrics to state that there

was any connection to photochemistry evident in this data set. I would need to see this suggestion/discussion removed before recommending publication.

Technical Corrections

There are extra spaces after most of the references when they end a sentence. Please edit the Latex code that is causing that.

In a number of cases there are spaces between value and % symbols, beginning in the abstract.

Pg 2 Line 1: insert a space between In and 1966

Pg 2 line 18: omit comma after "found that"

---

## Author Comment (AC2) · 8 May 2018

**The current paper reports organic concentration and speciation of aerosol organic functional groups during AWARE. There is a lack of measurements in the study region and these measurements are very timely. However, the paper is poorly written and it is not recommended for publication in ACP. A more specialized lower impact factor journal is recommended.**

We thank the referee for reviewing the manuscript and recognizing the importance of our unique, first-of-their-kind, yearlong measurements of organic aerosol mass and functional group composition. **(Page and line numbers in this response reference the location in the discussion paper where the text is inserted. The revised manuscript will note these revisions separately with tracked changes when it is posted.)**

Our findings are appropriate for ACP rather than a lower-impact journal because they meet the journal requirements, which are: "Research articles report substantial new results and conclusions from scientific investigations of atmospheric properties, and processes within the scope of the journal. Please note that the journal scope is focused on studies with general implications for atmospheric science rather than investigations which are primarily of local interest."(https://www.atmospheric-chemistry-and-physics.net/about/manuscript_types.html)

Specifically the atmospheric properties investigated in this manuscript were organic aerosol composition and mass emitted from marine and sea bird sources in Antarctica. Three main results are

1) OM and PM concentration were 5-150 times higher during summer than winter, with summer OM mostly from sea bird sources.
2) Summertime OM has amide group characteristic of seabird emissions.
3) Carboxylic acid group from natural sources were from secondary formation pathways.

These results are new because there are no published reports of any of these results. While Result (1) is consistent with seasonal trends for CN (Hogan and Barnard, 1978; Parungo et al., 1981; Bodhaine, 1983; Gras, 1993; Kim et al., 2017) and PM (Wagenbach et al., 1998; Jourdain and Legrand, 2002; Weller and Wagenbach, 2007; Udisti et al., 2012; Legrand et al., 2017a; Legrand et al., 2017b; Asmi et al., 2018) , prior results did not include OM. Similarly Result (2) is consistent with reports attributing some CHN and CHNO fragments to seabirds (Schmale et al., 2013) , but our results are the first to provide their contribution to PM and summertime maximum. Result (3) is consistent with observations of marine SOA (Frossard et al., 2014) but new in that this is the first study showing a coastal (i.e. continental not marine) source of SOA in Antarctica. These results are substantial because there are no other measurements of seasonal OM concentration in all of Antarctica, which makes calculation of radiative fluxes in this entire region very uncertain. In addition there is no prior evidence of an OM functional group marker for seabird sources, which has prevented the identification of the

contribution of seabird sources worldwide.  Finally, the lack of characterization of secondary particle sources in Antarctica means that this important pathway has not been included in models of aerosol formation and growth over Antarctica.

The reasons why this study is within the scope of the journal are:

1) The following past ACP publications of Antarctic aerosol measurements show that this topic is well within the scope of the journal: Giordano et al., 2017; Asmi et al., 2010; O'Shea et al., 2017; Legrand et al., 2017a; Legrand et al., 2017b; Hara et al., 2014; Hara et al., 2013.

2) Two specific ACP publications with results similar to ours are described below:
   a. Legrand et al. (2017a, b) reported multiple year size-segregated aerosol composition in central Antarctica (Concordia) and found chloride depletion as well as different seasonal patterns of nssSO$_4$ and MSA characterizing to particles in central Antarctica. Our year-round measurements at McMurdo Station has a similar scope to these two publications except that we measured organic functional groups in addition to inorganics and that our main results are about OM rather than sulfate.
   b. Giordano et al. (2017) reported aerosol measurements by AMS near McMurdo Station during October–December 2014 and August-October 2015 with source apportionment by PMF. The study found that sulfate aerosol mode was a large fraction of non-refractory submicron particle mass, even though the mass concentrations were not corrected for collection efficiency in two non-consecutive summer and spring months. Our year-round measurements at McMurdo Station have a similar scope to this publication except that we measured by FTIR and XRF rather than AMS and that our main results are about OM rather than sulfate.

3) Our study has general implications rather than just local interest for the following reasons:
   a. Because there are so few other measurements on the entire Antarctic continent, the closest other station with aerosol measurements is 300 km away.  In fact, McMurdo Station is one of the two sites that have published aerosol measurements starting in 1968, with the other one being the Amundsen Scott Station at the South Pole. The site has at least 10 publications describing aerosol measurements over the past 50 years, most of which were limited to summer (Cadle et al., 1968; Warburton, 1973; Ondov et al., 1973; Hogan, 1975; Hofmann, 1988; Hansen et al., 2001; Mazzera et al., 2001a; Mazzera et al., 2001b; Giordano et al., 2017; Kalnajs et al., 2013; Khan et al., 2018) .   Moreover McMurdo is the only station in the Ross Ice Shelf region.
   b. Marine and seabird sources are common along the coastlines of the continent of Antarctica, so the results from this study apply to much of coastal Antarctica which covers hundreds of square miles.

c. Specifically for the Crozier colony, we estimate the plume from seabirds to increase OM by 20 times for an area that covers 1000 km downwind with an average width of 50 km. This calculation is consistent with the 4 month summertime seabird-related OM concentrations of 0.16 $\mu$g m$^{-3}$ compared to winter of 0.001 $\mu$g m$^{-3}$ at McMurdo, which is >70 km downwind.

d. Since this increased OM causes submicron PM concentrations to increase 20% because of seabird emissions for such a large area, the direct and indirect effects of aerosols will be increased in these regions. For example, in the Arctic, the indirect effect of seabirds was shown to change by -0.5 W m$^{-2}$ pan-Arctic radiative forcing averaged over the 14,000,000 km$^2$ of the Arctic Ocean (Croft et al. (2016) .

We recognize the referee's concern that the manuscript is poorly written but we note that Referee #3 describes the work as well written. Nonetheless we recognize that there are always ways to improve written communication that is intended for diverse audiences so, in accordance with the more specific suggestions of Referee #1, we have improved the manuscript by the reorganization of the introduction, expansion of the discussion of relevant literature, improvements in transitions between topics, clarification of the language, and minor grammatical corrections, as described below:

1. Reorganization of the introduction:
a) Moved season definition and seasonal meteorological conditions from P4 Line 25 to section 2 P3 Line 11.
b) Moved description of PMF and K-means clustering operation to Method P3 Line 25 P5 Line15 (as described in response to Referee #1).
c) Improved organization of the fourth paragraph in section 3 P5 Line 23-33 (as described in response to Referee #1).

2. Improvements in transitions between topics (as described in response to Referee #1):
a) Rephrased P1 Line 30-31.
b) Rephrased P3 Line 9.
c) Rephrased P5 Line 1-22.

3. Expansion of the discussion of relevant literature (as described in response to Referee #1):
a) Added discussion of source of sea salt aerosols at P6 Line 6.
b) Added comparison to Arctic OM at P6 Line10.
c) Added summary of reasons for attribution of FTIR PMF factor to seabird and marine sources in at P8 Line 6.

4. Clarification of the language and methods.
a) Added more detailed about the measurement setup. (P3 Line 20) (as described in response to Referee #1).
b) Clarified OM calculation and detection limit for FTIR measurement (P4 Line 3 and P7 Line 19). (as described in response to Referee #1).

c) Clarified why the CN concentrations that remain after SLCE (spikes) are removed are considered representative of the natural background rather than local pollution from McMurdo (P4 Line 25) (as described in response to Referee #1).
d) Clarified the marine and FFC contributions to amine (P7 Line 22). (as described in response to Referee #1).
e) Changed units from imperial to metric "... less than 100km downwind..." (P7 Line 30)

5. Minor and grammatical corrections:
a) Rephrased "... modeling estimated that.." to "... modeling was used to estimate that... " (P2 Line 9).
b) Rephrased "Biological emissions from marine sulfate sources" to "Sulfate from marine biological... " (P2 Line 20).
c) Rephrased "The organic composition of particles in marine and Arctic regions..." to "For comparison, in marine and Arctic regions, the organic composition of particles..." (P2 Line 22).
d) Corrected abbreviation from "CNH and CNHO" to "CHN and CHNO" (P2 Line 32).
e) Added description of location of Cape Crozier (P7 Line 33).
f) Rephrased "more regionally-representative patterns " to "..regionally-representative or "background" concentrations " (P8 Line 21).
g) Rephrased sentence "...the FFC factor had a higher concentration than M&S in winter but the concentrations were so low that the quantification of the M&S factor in winter is very uncertain." (P8 Line 31).
h) Fixed capitalization"... M&S factor..." (P9 Line 2).
i) Rephrased "seabird" to "seabird-related" (P9 Line 3).

Major issues:
- The paper does not present any introduction of the aerosol composition in Antarctica. Mostly, the introduction seems a report of the history of a specific station.

The Introduction of the original posted ACPD paper summarizes the measured aerosol composition and concentration at McMurdo Station as well as the few existing hygroscopicity and organic aerosol measurements in all of Antarctica. We agree that the Introduction could be expanded to provide
   1) a description of the paucity of nearby stations and the lack of long-term (more than one year) organics measurements (see item a) below).
   2) a more thorough review of the limited study of seasonal trends in Antarctica (see items b) and c) below).
   3) a more thorough review of the detailed OM measurements in Antarctica (see items d) and e) below).

These changes are itemized below:
a) Start of the second paragraph:
"Since McMurdo Station is the only site with measurements of PM, EC, OC, number concentrations that is within 300 km of the Ross Ice Shelf (which covers an area of more than 500,000 km$^2$). Furthermore, the station is unique in that McMurdo Station is one of the two sites that have published aerosol measurements starting in 1968, with the other

one being the Amundsen Scott Station at the South Pole. The site has at least 10 publications describing aerosol measurements over the past 50 years, most of which were limited to summer (Cadle et al., 1968; Warburton, 1973; Ondov et al., 1973; Hogan, 1975; Hofmann, 1988; Hansen et al., 2001; Mazzera et al., 2001a; Mazzera et al., 2001b; Giordano et al., 2017; Kalnajs et al., 2013; Khan et al., 2018) . No stations in Antarctica measured inorganic chemical composition year-round until 1978 (Parungo et al., 1981) , and none have measured year-round organic components. " (P2 Line3)

b) End of the second paragraph:
"Many measurement campaigns were limited to austral summer months because of restrictions on access (Cadle et al., 1968; Ondov et al., 1973; Warburton, 1973) and so lack information on seasonal changes." (P2 Line21)

c) After the second paragraph:
"The few year-round aerosol concentration and composition measurements in Antarctica were collected at several sites in coastal Antarctica (all of which are more than 1500 km from McMurdo Station) (Hara et al., 2005; Wagenbach et al., 1998; Jourdain and Legrand, 2002; Gras, 1993; Hara et al., 2004; Hara et al., 2010; Weller et al., 2013; Minikin et al., 1998; Read et al., 2008) and at several sites on the Antarctic Peninsula (more than 3000 km from McMurdo Station) (Asmi et al., 2018; Mishra et al., 2004; Kim et al., 2017; Saxena and Ruggiero, 1990; Savoie et al., 1993; Loureiro et al., 1992) , as well as at the South Pole (more than 1000 km from McMurdo Station) (Hansen et al., 1988; Bodhaine et al., 1986; Harder et al., 2000; Parungo et al., 1981; Bodhaine, 1983; Hogan and Barnard, 1978) and at Dome C (more than 1000 km from McMurdo Station) (Legrand et al., 2017b; Legrand et al., 2017a; Udisti et al., 2012) . At the South Pole, aerosol particle number concentration ranged from 10 to 30 cm$^{-3}$ in winter and 100 to 300 cm$^{-3}$ in summer (Bodhaine, 1983; Parungo et al., 1981; Hogan and Barnard, 1978) . This low winter and high summer seasonal difference has been observed also at coastal Antarctic sites, but the average concentrations were typically higher with summertime concentrations ranging from 300 to 2000 cm$^{-3}$ and wintertime concentrations from 10 to 200 cm$^{-3}$ (Kim et al., 2017; Gras, 1993) . Consistent with this seasonal difference in particle number concentrations, most summertime non-sea salt sulfate mass concentrations were at least 5 times higher than winter concentrations (Jourdain and Legrand, 2002; Weller and Wagenbach, 2007; Udisti et al., 2012; Legrand et al., 2017a; Asmi et al., 2018) , likely because of the contributions from biogenic DMS emissions from the surrounding Southern Ocean. However, most sea salt aerosols had wintertime maximum concentrations with more than two times more Na$^{+}$ mass concentrations in winter than summer (Parungo et al., 1981; Wagenbach et al., 1998; Jourdain and Legrand, 2002; Weller and Wagenbach, 2007; Jourdain et al., 2008; Udisti et al., 2012; Legrand et al., 2017b; Legrand et al., 2017a; Asmi et al., 2018) " (P3 Line1)

d) After the old P2 Line 22 line:
Three sentences about organic nitrogen were revised as suggested by Referee #1. (as described in response to Referee #1).

e) Before the old P2 line 29:

"Sugar, levoglucosan, phenols and anthropogenic persistent organic compounds were measured in ambient aerosols at Mario Zucchelli Station and Concordia Station (Zangrando et al., 2016; Barbaro et al., 2016; Barbaro et al., 2017; Barbaro et al., 2015) Carboxylic acids with low molecular weights were also measured at Mario Zucchelli Station, Concordia Station, and Dumont d'Urville (Barbaro et al., 2017; Legrand et al., 2012)

- Scientific questions and objectives are really not presented, neither discussed or summarized.

The major scientific questions and the corresponding three main results are presented, discussed, and summarized in the discussion paper at the following lines:
1) The question of how the concentration and composition of OM varies seasonally is presented on P1 at Line 20 in the abstract and on P3 at Line 3-4 in the introduction; it is discussed on P7 at Lines 10-21 in section 4; and it is summarized on P8 at Line27-28 in the conclusions.

2) The question of what the contributions from natural sources of OM are is presented on P1 at Line 21 and on P1 at Line 25 in the abstract and on P3 at Line 5-7 in the introduction; it is discussed on P7 at Line 10-35 and on P8 at Line 1-6 in section 4; and it is summarized on P8 at Line 29-31 and on P9 at Line 1-3 in the conclusions.

3) The question of whether or not there is a secondary pathway that contributes to OM formation is presented on P1 at Line 25-26 in the abstract and on P3 at Line 8-9 in the introduction; it is discussed on P8 at Line 7-17 in section 4; and it is summarized on P9 at Line3-5 in the conclusions.

In addition, we have now revised part of the abstract and introduction so that these scientific questions can be more easily identified by readers as follows:

In abstract: "Observations of the organic components of the natural aerosol are scarce in Antarctica, which prevent us from better understanding natural aerosols and their connection to seasonal and spatial patterns of cloud albedo in the region."

In the introduction: "This manuscript characterizes the sources of organic aerosol across four seasons in Antarctica. Dust, sea salt, and non-sea salt sulfate mass concentrations measured by XRF are used to separate the seasonal contributions to inorganic particle components.  Seasonal patterns of natural marine and coastal-sourced organic aerosol are identified from the functional groups after separation of local emissions. " (P3 Line9)

- After reading a paper a number of times, and looking at the figures, one can argue the main results are the impact of sea birds and marine sources in Antarctica. Not surprisingly at all, the reader does not understand if this is simply a bad measurement site (bad luck) or if the study has any implication.

We agree with the Reviewer that one of the three main results of this paper is the contribution of seabird emissions to summertime submicron particles. We further note that this result is important because of the reasons noted above for why McMurdo Station is not only a good and representative sampling site but also the home of one of the earliest aerosol measurements on the continent.

We have added the following text to the discussion to highlight the potential regional implications of seabird emissions based on results for the Arctic:
"The emissions from seabirds have significant regional implications in polar areas due to because of their large population and wide distribution (Croft et al., 2016; Riddick et al., 2012)  Chemical transport model simulations suggest that emissions of reduced nitrogen from seabirds in the Arctic could significantly increase aerosol particle formation, and in turn cloud droplet number concentration and cloud albedo, yielding as much as -0.5 W m$^{-2}$ radiative forcing averaged over the 14,000,000 km$^2$ of the Arctic Ocean (Croft et al., 2016) ."

In addition, to make a simple approximation of the impact of seabird plumes in Antarctica (see Appendix), we use a simple Gaussian dispersion model to estimate that the 80,000 Cape Crozier seabird breeding pairs will affect an area of approximately 50,000 km$^2$ by increasing aerosol optical by 40% (from 0.010 to 0.014). Scaling this up to all Adelie penguins in Antarctica (2 to 3 million breeding pairs) and assuming not overlapping plumes of similar dispersion, we estimate that as much as 4,000,000 km$^2$ could be affected by the approximately 200 seabird colonies along the Antarctic coastline (Borowicz et al., 2018; Ainley, 2002; Knox, 2006) .

I am afraid I cannot be any positive at this stage.

We are very sorry that the referee "cannot be any positive" at this stage. The coauthors on this manuscript request that the Referee reconsider his/her opinion based on the revisions presented here. Specifically we argue that the manuscript is appropriate for ACP since we are presenting substantial and new results with general implications. We thank the referee for reconsideration and we are happy to provide additional responses and clarification if needed.

**Appendix**

**1. Plume Dispersion:**
The following equation is used to calculate the concentration (C) from dispersion of the plume from the seabird colony emissions:

$$C(x, y, z; H) = \frac{Q}{\sqrt{2\pi}LU\sigma_y} exp[-\frac{1}{2}(\frac{y}{\sigma_y})^2] \qquad \text{(Turner, 1994)}$$

Table A1. Parameters for plume calculation.

| Parameters | Description | Value |
|---|---|---|
| x | horizontal distance along the wind direction | calculated |
| y | horizontal distance normal to the wind direction | calculated |
| z | vertical distance from the source | calculated |
| L | boundary layer height | 400 m |
| H | height of emission source | 0 m |
| Q | emission strength of the point source | $6 \times 10^6 \ \mu g \ s^{-1}$ |
| $\sigma_y$* | dispersion coefficient in y direction, determined from measured atmospheric conditions | 814 m |

*$\sigma_y$ was determined for stability class C for the average wind speed of 6 m s$^{-1}$ and radiation measured at McMurdo Station (Martin, 1976) . (Pasquill, 1961)

This is a simplified Gaussian plume dispersion model for capped inversion. We assumed that emission was from a point source with zero elevation at Crozier and ignored particle sinks during transport. Boundary layer height was calculated by the ARM standard method (Liu and Liang, 2010) from sounding profiles measured during AWARE, giving a summertime average boundary layer height of 400 m (Table A1). We used a constant wind speed of 6 m s$^{-1}$ and direction from Cape Crozier to McMurdo Station. Fitting this equatio to the observed average summer concentration of 0.16 $\mu g$ m$^{-3}$ OM at McMurdo Station gives Q of $6 \times 10^6$ $\mu g$ s$^{-1}$ for OM. To estimate the ammonium contribution to PM, the 80,000 breeding pairs at Crozier (Lyver et al., 2014)  have an ammonia emission strength of  $2 \times 10^6$ $\mu g$ s$^{-1}$ from equation (2) (Riddick et al., 2012) with the estimated coefficients for the nitrogen metabolism in that study. If all of the ammonia is taken up by submicron particles then the ammonium concentration from seabirds at McMurdo Station is estimated to be 0.05 $\mu g$ m$^{-3}$, which is reasonable given the measured submicron sulfate mass concentration. Using the Q calculated for OM and defining the plume to be the area for which OM exceeds 0.02 $\mu g$ m$^{-3}$ (20 times higher than the wintertime natural OM average of 0.001 $\mu g$ m$^{-3}$), we find the plume has an estimated length of 1000 km with an average width of 50 km, giving an area of 50,000 km$^2$ with average OM concentration of 0.04 $\mu g$ m$^{-3}$. Including the ammonium concentration for the same plume area adds an additional 0.01$\mu g$ m$^{-3}$ to PM1, making a total of 0.05 $\mu g$ m$^{-3}$ from seabirds for the Crozier plume.

**2. Extinction:**

The background AOD at 500 nm (Cimel Sunphotometer) was determined to be 0.05 by using the average of minimum values from 18 November 2015 to 29 February 2016 (https://www.arm.gov/research/campaigns/amf2015aware).  This number is consistent with measurements of median aerosol optical depth (500 nm) of total aerosol, which ranged from 0.02 to 0.06 at 9 stations across Antarctica (Tomasi et al., 2007) . To estimate the fraction of AOD that is from submicron scattering, we scale the AOD by the ratio of PM$_1$ to PM$_{10}$ scattering (2.3/3.3) to get 0.035 (Table A2).  Of this, we can estimate the fraction associated with seabirds by scaling by the ratio of seabird-related PM$_1$ to PM$_1$ (0.21/0.52) to get 0.014 at McMurdo Station. The seabird-related part of

AOD for the entire plume spread over 50000 km$^2$ is then one-fourth of this (0.05/0.20), at 0.004.

If the 80,000 seabird breeding pairs at Cape Crozier affect an area of approximately 50,000 km$^2$ by increasing aerosol optical by 40% (from 0.010 to 0.014), then the 2 to 3 million penguin breeding pairs in Antarctica (assuming not overlapping plumes of similar dispersion) will affect as much as 4,000,000 km$^2$ (Borowicz et al., 2018; Ainley, 2002; Knox, 2006) (Table A3).

Table A2. Estimate of seabird-related AOD from the Crozier plume.

| Quantity | McMurdo Station (70 km downwind) | Crozier Plume (Averaged over 50000 km$^2$) |
|---|---|---|
| PM$_1$ scattering coefficient | 2.3 1/Mm | |
| PM$_{10}$ scattering coefficient | 3.3 1/Mm | |
| PM$_1$ (seabird-related) | 0.21 $\mu$g m$^{-3}$ | 0.05 $\mu$g m$^{-3}$ |
| PM$_1$* | 0.52 $\mu$g m$^{-3}$ | 0.20 $\mu$g m$^{-3}$ |
| AOD total | 0.050 | - |
| AOD submicron | 0.035 | |
| AOD submicron seabird | 0.014 | 0.004 |

*PM$_1$ is calculated as the sum of the measured OM, NSS sulfate, and sea salt, plus the ammonium mass concentration from the Crozier plume calculated from the Riddick et al. (2012) source and diluted to the distance of McMurdo Station (70 km).

Table A3. Estimate of area influenced by Adelie penguin emission.

| | Penguin Breeding Pairs | Potential Plume Area |
|---|---|---|
| Cape Crozier | 80,000 | 50,000 km$^2$ |
| Antarctica | 2 million to 3 million | 1 to 4 million km$^2$ |

---

## Author Comment (AC3) · 26 May 2018

Comments on Liu 2018 Antarctica aerosol

**General Comments**

**This paper covers a year's worth of organic aerosol measurements in a region that is rarely sampled and sheds valuable insight into the chemical composition of Antarctic aerosol. The paper is well-written and the figures and tables are clear and legible. Aside from minor corrections and qualifications, I have just one concern with the current form of the manuscript. The authors claim a connection between carboxylic acid variability and downwelling radiation that, for reasons I describe below, is misleading. Unless the authors can clarify and justify this correlation, I would recommend that the discussion of that connection be omitted. Otherwise the paper needs only minor revision before it is suitable for publication. Given that this aspect may take more work to revise, I am selecting major revision in the online evaluation. However, the changes should not be overly burdensome.**

The authors thank the referee for the review. We have addressed the concern about the carboxylic acid disscussion and have clarified that point below. We also addressed other minor comments and concerns, and these have improved the manuscript as itemized below. **(Page and line numbers in this response reference the location in the discussion paper where the text is inserted. The revised manuscript will note these revisions separately with tracked changes when it is posted.)**

Specific Comments

Pg 2 line 22: please give a value range for the "high fraction" of OH observed previously

We added the range requested as follows: "...high fraction of hydroxyl group (61% of OM for the North Atlantic and 47% of OM for the Arctic)..." (P2 line22)

Pg 2 line 25: similarly, what range of ON mass fractions have been observed? Is it a minor component, or major? Or highly variable?

We added the range as follows: "Organic nitrogen has also been identified as a tracer component (0.02 to 10 ng m$^{-3}$)... " (P2, Line 25)

Pg 5 line 5: I don't see SLCE defined before it is use as an abbreviation

We thank the referee for noticing this and have corrected this as follows: " ...which we attributed to short-term local contamination events (SLCE) (Figure S1)." (P4 Line 29)

Pg 7 line 4: I'm unclear as to what the "factors identified as urban combustion emissions" is that correlates to the FFC factor. Do the authors mean "factor spectra"? Like, other FTIR PMF spectra? Please clarify.

We have clarified this as follows: "... with factor spectra identified previously..." (P7 Line 4)

Figure 6 and discussion on page 8 (and in conclusion/abstract): Drawing any relationship about photochemistry from the correlation between M&S carboxylic acid and DWR is misleading. The observed correlation, as I understand it, is simply the correlation between the M&S factor strength and downwelling radiation time series, since the carboxylic acid attributed to the M&S factor is always the same fraction of the factor (given in Fig 4), and so varies only as the strength of that factor. The same correlation coefficient (r) would be obtained for any of the functional groups present in the M&S factor and for the factor as a whole, as correlation coefficients do not change with addition/subtraction or multiplication by constants to the vectors being compare. Further, the downwelling radiation is varying only because of the season change (Fig 1) and the strength of the M&S factor, associated with the Adelie penguins, is also due to seasonal migration, so the observed correlation to downwelling is really just a product of the M&S factor and downwelling both having season characteristics. The authors would have to do more analysis and include other metrics to state that there was any connection to photochemistry evident in this data set. I would need to see this suggestion/discussion removed before recommending publication.

We thank the referee for this comment. We agree that the correlation to radiation itself does not indicate secondary aerosol formation given the overlap of the seasonality of the emissions and radiation. We have revised this discussion to note that carboxylic acids have been shown to be from secondary aerosol formation and that in this study they are associated with the natural seabird emission source. We have revised this section in the Discussion, in part in response to Referee #1:

[FROM RESPONSE TO REFEREE #1]"The measured acid group concentration is likely to be a secondary aerosol contribution since photochemical oxidation has been shown to form highly oxidized molecules including carboxylic acids by photochemical reactions (Alves and Pio, 2005; Charbouillot et al., 2012) . Acids are also present in trace amounts in seawater (Gagosian and Stuermer, 1977; Kawamura and Gagosian, 1987) , but the higher concentrations measured here are likely to only be explained by secondary processes. The carboxylic acid group mass concentration that was associated with the M&S factor was correlated moderately to downwelling shortwave irradiance (r=0.75, Figure 6), supporting the idea that the carboxylic acid group mass was from photochemical reactions." (P8 Line 7)

[FROM RESPONSE TO REFEREE #1]"Carboxylic acid group mass fractions have also been identified as secondary photochemical products based on their correlation to solar radiation in clean, open-ocean conditions (Frossard et al., 2014) . However, since the seabird emissions were only high in summer when radiation was also generally high, the correlation to radiation does not provide evidence of photochemical contributions in this case. Interestingly, the carboxylic acid group associated with the FFC factor had no correlation (r= 0.09) to downwelling shortwave irradiance. This difference may be because the local emissions from McMurdo Station facilities reached the Cosray site in less than 5 min (since McMurdo Station was 2 km away and wind speeds were 6 m s-1 on average) making them essentially "fresh" primary particles, whereas those from the large upwind penguin colony took 6 hr (since Cape Crozier was 100 km away and wind speeds were 6 m s-1 on average) to reach the site giving them approximately 50 times

more time for photochemical reactions leading to SOA production. It is also possible that the anthropogenic gas-phase precursor emissions had lower SOA acid yields but there is little evidence to support this(Rickard et al., 2010; Wyche et al., 2009; McNeill, 2015). The source of the vapor-phase organic precursors of the summer seabird acid groups is not known, but given their substantial contribution to mass is worthy of further investigation." (P8 Line 13)

In addition, we have revised the following sentences in these two paragraphs to better support this point:

"The carboxylic acid group mass concentration that was associated with the M&S factor was correlated moderately to downwelling shortwave irradiance (r=0.75, Figure 6). "(P8 Line 7)

 "The measured acid group concentration is likely to be a secondary aerosol contribution since photochemical oxidation has been shown to form highly oxidized molecules, including carboxylic acids (Xu et al., 2013;Barbaro et al., 2017;Kawamura and Gagosian, 1987;Sax et al., 2005;Charbouillot et al., 2012;Alves and Pio, 2005;Claeys et al., 2007;Alfarra et al., 2006;Stephanou and Stratigakis, 1993)."(P8 Line 13)

We also revised the Abstract and Conclusion as follows:

Rephrased from "...sources were correlated to incoming solar radiation, indicating that some OM formed by secondary pathways." to "Carboxylic acid group contributions were high in summer and associated with natural sources, likely forming by secondary reactions. "(P1 Line 25)

Rephrased from "was well correlated to downwelling shortwave irradiance (r=0.69) and was likely from secondary products of photochemical reactions" to "...was high in summer and was likely from secondary products of photochemical reactions... " (P9 Line 4)

Technical Corrections
There are extra spaces after most of the references when they end a sentence. Please edit the Latex code that is causing that.
The format is corrected as suggested.

In a number of cases there are spaces between value and % symbols, beginning in the abstract.
We thank the referee for noticing them. The format is corrected as suggested.

Pg 2 Line 1: insert a space between In and 1966; Pg 2 line 18: omit comma after "found that"
We have corrected these two typos as suggested.

**References**

Alfarra, M. R., Paulsen, D., Gysel, M., Garforth, A. A., Dommen, J., Prévôt, A. S., Worsnop, D. R., Baltensperger, U., and Coe, H.: A mass spectrometric study of secondary organic aerosols formed from the photooxidation of anthropogenic and biogenic precursors in a reaction chamber, Atmospheric Chemistry and Physics, 6, 5279-5293, 2006.

Alves, C. A., and Pio, C. A.: Secondary organic compounds in atmospheric aerosols: Speciation and formation mechanisms, Journal of the Brazilian Chemical Society, 16, 1017-1029, 10.1590/s0103-50532005000600020, 2005.

Barbaro, E., Padoan, S., Kirchgeorg, T., Zangrando, R., Toscano, G., Barbante, C., and Gambaro, A.: Particle size distribution of inorganic and organic ions in coastal and inland Antarctic aerosol, Environmental Science and Pollution Research, 24, 2724-2733, 10.1007/s11356-016-8042-x, 2017.

Charbouillot, T., Gorini, S., Voyard, G., Parazols, M., Brigante, M., Deguillaume, L., Delort, A. M., and Mailhot, G.: Mechanism of carboxylic acid photooxidation in atmospheric aqueous phase: Formation, fate and reactivity, Atmospheric Environment, 56, 1-8, 10.1016/j.atmosenv.2012.03.079, 2012.

Claeys, M., Szmigielski, R., Kourtchev, I., Van der Veken, P., Vermeylen, R., Maenhaut, W., Jaoui, M., Kleindienst, T. E., Lewandowski, M., and Offenberg, J. H.: Hydroxydicarboxylic acids: markers for secondary organic aerosol from the photooxidation of $\alpha$-pinene, Environmental science & technology, 41, 1628-1634, 2007.

Guzman-Morales, J., Frossard, A. A., Corrigan, A. L., Russell, L. M., Liu, S., Takahama, S., Taylor, J. W., Allan, J., Coe, H., Zhao, Y., and Goldstein, A. H.: Estimated contributions of primary and secondary organic aerosol from fossil fuel combustion during the CalNex and Cal-Mex campaigns, Atmospheric Environment, 88, 330-340, 10.1016/j.atmosenv.2013.08.047, 2014.

Kawamura, K., and Gagosian, R. B.: Implications of omega-oxocarboxylic acids in the remote marine atmosphere for photooxidation of unsaturated fatty acids, Nature, 325, 330-332, 10.1038/325330a0, 1987.

Price, D. J., Chen, C.-L., Russell, L. M., Lamjiri, M. A., Betha, R., Sanchez, K., Liu, J., Lee, A. K. Y., and Cocker, D. R.: More unsaturated, cooking-type hydrocarbon-like organic aerosol particle emissions from renewable diesel compared to ultra low sulfur diesel in at-sea operations of a research vessel, Aerosol Science and Technology, 51, 135-146, 10.1080/02786826.2016.1238033, 2017.

Russell, L. M., Hawkins, L. N., Frossard, A. A., Quinn, P. K., and Bates, T. S.: Carbohydrate-like composition of submicron atmospheric particles and their production from ocean bubble bursting, Proceedings of the National Academy of Sciences of the United States of America, 107, 6652-6657, 10.1073/pnas.0908905107, 2010.

Sax, M., Zenobi, R., Baltensperger, U., and Kalberer, M.: Time resolved infrared spectroscopic analysis of aerosol formed by photo-oxidation of 1,3,5-trimethylbenzene and alpha-pinene, Aerosol Science and Technology, 39, 822-830, 10.1080/02786820500257859, 2005.

Stephanou, E. G., and Stratigakis, N.: Oxocarboxylic and. alpha.,. omega.-dicarboxylic acids: photooxidation products of biogenic unsaturated fatty acids present in urban aerosols, Environmental science & technology, 27, 1403-1407, 1993.

Xu, G. J., Gao, Y., Lin, Q., Li, W., and Chen, L. Q.: Characteristics of water-soluble inorganic and organic ions in aerosols over the Southern Ocean and coastal East Antarctica during austral summer, Journal of Geophysical Research-Atmospheres, 118, 13303-13318, 10.1002/2013jd019496, 2013.